# APOE4 exacerbates synapse loss and neurodegeneration in Alzheimer's disease patient iPSC-derived cerebral organoids

Jing Zhao[1,2], Yuan Fu[1], Yu Yamazaki [1], Yingxue Ren[3], Mary D. Davis[1,2], Chia-Chen Liu [1], Wenyan Lu[1,2], Xue Wang[3], Kai Chen[1], Yesesri Cherukuri[3], Lin Jia[1], Yuka A. Martens [1], Lucy Job[1,2], Francis Shue [1], Thanh Thanh Nguyen [1], Steven G. Younkin[1], Neill R. Graff-Radford[4], Zbigniew K. Wszolek[4], David A. Brafman[5], Yan W. Asmann[3], Nilüfer Ertekin-Taner[1,4], Takahisa Kanekiyo[1,2] & Guojun Bu [1,2✉]

*APOE4* is the strongest genetic risk factor associated with late-onset Alzheimer's disease (AD). To address the underlying mechanism, we develop cerebral organoid models using induced pluripotent stem cells (iPSCs) with *APOE* ε3/ε3 or ε4/ε4 genotype from individuals with either normal cognition or AD dementia. Cerebral organoids from AD patients carrying *APOE* ε4/ε4 show greater apoptosis and decreased synaptic integrity. While AD patient-derived cerebral organoids have increased levels of Aβ and phosphorylated tau compared to healthy subject-derived cerebral organoids, *APOE4* exacerbates tau pathology in both healthy subject-derived and AD patient-derived organoids. Transcriptomics analysis by RNA-sequencing reveals that cerebral organoids from AD patients are associated with an enhancement of stress granules and disrupted RNA metabolism. Importantly, isogenic conversion of *APOE4* to *APOE3* attenuates the *APOE4*-related phenotypes in cerebral organoids from AD patients. Together, our study using human iPSC-organoids recapitulates *APOE4*-related phenotypes and suggests *APOE4*-related degenerative pathways contributing to AD pathogenesis.

[1] Department of Neuroscience, Mayo Clinic, Jacksonville, FL 32224, USA. [2] Center for Regenerative Medicine, Neuroregeneration Lab, Mayo Clinic, Jacksonville, FL 32224, USA. [3] Department of Health Sciences Research, Mayo Clinic, Jacksonville, FL 32224, USA. [4] Department of Neurology, Mayo Clinic, Jacksonville, FL 32224, USA. [5] School of Biological and Health Systems Engineering, Arizona State University, Tempe, AZ 85287, USA. ✉email: bu.guojun@mayo.edu

Alzheimer's disease (AD) characterized by progressive neurodegeneration is the most common form of dementia[1]. While formations of senile plaques and neurofibrillary tangles (NFTs) are two major neuropathological hallmarks in AD, these pathologies composed of amyloid-β peptides (Aβ) and phosphorylated tau often precede the onset of symptomatic dementia by decades[2,3].

The ε4 allele of *APOE* gene (*APOE4*) is the strongest genetic risk factor for AD among the three polymorphic alleles (*APOE2*, *APOE3*, and *APOE4*)[4,5]. Apolipoprotein E (apoE) is produced primarily by astrocytes in the central nervous system as a carrier of cholesterol and other lipids to support membrane homeostasis, synaptic integrity, and injury repair[6,7]. Increasing evidences from animal models and postmortem human brains have shown that *APOE4* is associated with multiple aspects of AD pathogenesis[8]. In particular, mouse models carrying human *APOE4* have accelerated Aβ seeding and suppressed Aβ clearance, as well as disturbed synaptic plasticity and blood–brain barrier integrity[7,9]; however, whether these findings are translatable to humans is not clear. In addition, studies with human postmortem brains represent only the characteristics of end-stage disease[10–12]. Thus, there is an urgent need to establish human-relevant models to define *APOE4*-related pathogenic pathways in AD.

The induced pluripotent stem cells (iPSCs) derived from human somatic cells with AD-linked mutations or risk alleles are promising in vitro models by recapitulating the earliest molecular and pathological changes in age-related disorders[13–17]. Elevated levels of Aβ and phosphorylated tau, as well as increased cellular stress markers have been reported in iPSC-derived neurons from AD patients[14–16,18]. For example, Aβ oligomers accumulate in iPSC-derived neurons and astrocytes in cells from patients with a familial amyloid precursor protein (*APP)* p.E693Δ mutation and sporadic AD, leading to endoplasmic reticulum (ER) and oxidative stress[15]. In addition to altered APP processing, an increase in levels of total and phosphorylated tau is observed in iPSC-derived neurons from AD patients carrying the *APP* p.V717I mutation or *APOE4* risk allele[18,19]. However, the relevance of the observations from two-dimensional (2-D) cell cultures to AD is questionable as AD pathology is intricate and involves diverse cell types, tissue structures, and cellular pathways. Thus, in this study, we utilize a large number of human iPSC lines from healthy subjects and AD patients carrying *APOE* ε3/ε3 or ε4/ε4 genotype, and investigate AD-related phenotypes using the iPSC-derived three-dimensional (3-D) cerebral organoid model system, which is highly reminiscent of human brain structure with diverse cell types[20–22]. Here, we show that *APOE4* aggravates neurodegeneration in iPSC-derived cerebral organoids from AD patients; however, its effects on tauopathy are significant in both healthy subject-derived and AD patient-derived cerebral organoids. Interestingly, the levels of Aβ are increased in AD organoids independent of *APOE* genotype. Importantly, isogenic conversion of *APOE4* to *APOE3* attenuates AD-related phenotypes in iPSC-derived cerebral organoids. These findings reveal *APOE4*-related pathways in 3-D models that are directly relevant to humans.

## Results

**Characterization of human iPSC-derived cerebral organoids.** To study the effects of *APOE* genotype on AD-related pathways in a physiologically relevant environment, we generated 3-D cerebral organoid models using human iPSC lines from cognitively unimpaired individuals carrying *APOE* ε3/ε3 (Con-E3; $N = 5$), cognitively unimpaired individuals carrying *APOE* ε4/ε4 (Con-E4; $N = 5$), AD patients carrying *APOE* ε3/ε3 (AD-E3; $N = 5$) and AD patients carrying *APOE* ε4/ε4 (AD-E4; $N = 5$), collected from multiple sources (Supplementary Table 1).

Among them, 5 iPSCs lines were generated de novo from fibroblasts by transfecting with three episomal vectors encoding five transcription factors (*OCT3/4*, *SOX2*, *L-MYC*, *KLF4*, and *LIN28*) and p53 shRNA[23]. Expression of pluripotency stem cell-specific markers was confirmed by immunostainings for SSEA4, Nanog, and TRA-1-60 (Supplementary Fig. 1A). The pluripotency of the iPSC lines was also validated by their ability to differentiate into endodermal, mesodermal, and ectodermal origin cells upon immunostaining of Brachyury (Mesoderm marker), Sox17 (endoderm marker), and Nestin/Sox2 (ectoderm marker) (Supplementary Fig. 1B). The iPSC lines maintained a normal karyotype after reprogramming (Supplementary Fig. 1C). All other iPSC lines utilized in published studies[19,24–26] or from California Institute for Regenerative Medicine (CIRM) (FUJIFILM Cellular Dynamics, Inc.) have been fully validated and characterized previously.

To generate cerebral organoids from iPSCs with efficiency and reproducibility, we followed an optimized protocol developed by Stemcell Technologies. Cerebral organoid formation was initiated through an intermediate embryonic body (EB) formation step followed by expansion of neuroepithelia in a matrigel scaffold. On day 12, the iPSC-derived organoids were transferred to an orbital shaker in neuronal differentiation medium and maintained under rotary conditions for maturation (Fig. 1a). On week 4, cerebral organoids showed a predominantly dorsal forebrain region specification, containing fluid-filled ventricle-like structures aligned with Sox2-positive neural progenitors in a ventricular/subventricular-like zone (VZ/SVZ) and beta-tubulin III (Tuj1)-positive neuroblasts in an outer layer (Fig. 1b). A deep cortical layer marker Ctip2-positive neurons were detected as early as week 4 (Fig. 1c), whereas a superficial cortical layer marker Satb2-positive neurons emerged in the later stage at week 12 (Fig. 1d). These observations revealed the sequential emergence of different neuronal layers along with the differentiation, which is consistent with previous publication[20]. Since apoE is mainly produced by astrocytes in the brain[27], we assessed the presence of astrocytes at different time points by immunostaining for glial fibrillary acidic protein (GFAP). We found that small clusters of GFAP-positive astrocytes started to emerge in some VZ area at week 4. GFAP-positive astrocytes showed an immature morphology with short processes, which were separated from surrounding Tuj1-positive neuronal cells (Fig. 1e). At week 12 of differentiation, GFAP-positive astrocytes increased in number and migrated within the neuronal layers, displaying typical mature astrocyte morphology with long processes (Fig. 1f). Furthermore, to determine the influence of the technical difference to the differentiation of cerebral organoids across different rounds, we compared the levels of GFAP and Tuj1 in the cerebral organoids from two rounds of differentiation. We found no significant differences in the expression of GFAP and Tuj1 at week 12 between the first and second round of experiments (Supplementary Fig. 9). These results indicate the successful development of iPSCs to cerebral organoids with cortical like organization composed of abundant mature neurons and astrocytes.

**APOE4 and AD status exacerbate neurodegeneration.** With the establishment of the cerebral organoid culture system, we generated four groups of iPSC-derived cerebral organoids (Con-E3, Con-E4, AD-E3, and AD-E4). Cerebral organoid size was monitored and there were no evident differences observed among different groups at 2 and 12 weeks (Supplementary Fig. 2). To evaluate the effect of *APOE4* and disease status on neuronal apoptosis/degeneration in the cerebral organoids, cleaved caspase-3 (CASP3) was analyzed by immunostaining at week 12 (Fig. 2a). To avoid the possible influences of necrosis observed in

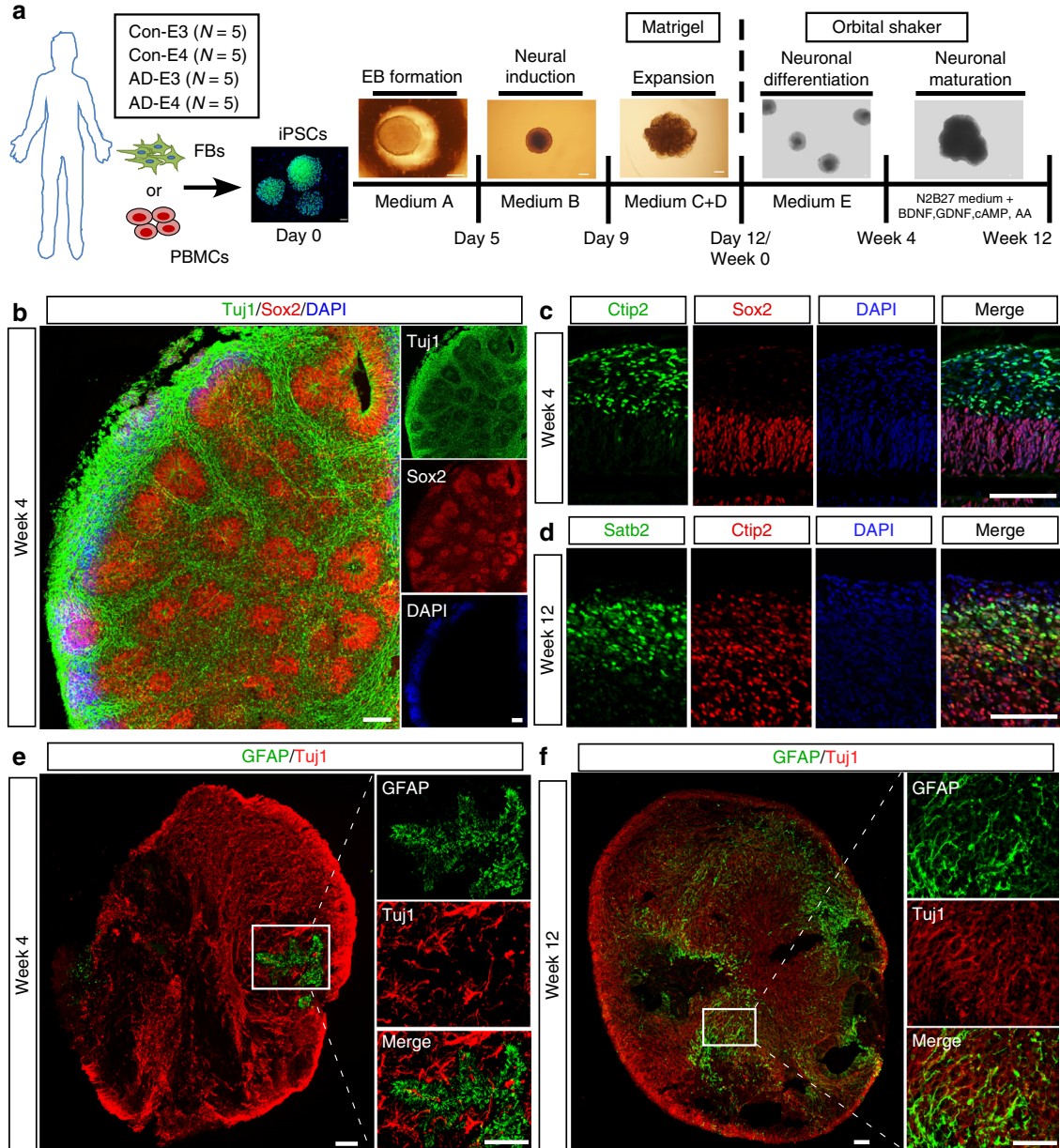

**Fig. 1 Generation and characterization of cerebral organoids from human iPSCs. a** A schematic overview of the procedures for generating cerebral organoids from human iPSCs using the STEMdiff™ Cerebral Organoid Kit. **b** Representative images of the ventricular zone (VZ)-like structure formed by new-born neurons (Tuj1, green) and neural progenitor cells (Sox2, red) in cerebral organoids at week 4 of differentiation. **c, d** Confocal images showed the cortical layer structure formation when immunostained for Ctip2 (deep cortical layer marker) and Satb2 (superficial cortical layer marker) at week 4 (**c**) and 12 (**d**), respectively. **e, f** Proliferation and migration of astrocytes within cerebral organoids; the differentiation pattern of astrocytes in organoids were monitored by GFAP immunostaining (astrocytic marker) at week 4 (**e**) and 12 (**f**), respectively. Scale bar: 100 μm.

the core portion of organoids, the immunoreactivity was measured only in the surface neuronal layers. We found the increased cleaved CASP3 immunoreactivity in the AD-E4 organoids compared to other groups with an interactive effect between *APOE4* and AD status (Fig. 2b). Consistently, western blotting also revealed higher cleaved CASP3/CASP3 ratio in AD-E4 organoids (Fig. 2c, d). These results indicate that *APOE4* and AD status synergistically exacerbate apoptosis in late stage of organoid development. Presynaptic synaptophysin and postsynaptic PSD95 were decreased in AD organoids groups compared to healthy subject-derived cerebral organoids, whereas no significant *APOE4* effect was observed (Fig. 2c, e, and f). In contrast, on week 4, reverse transcription-quantitative PCR (RT-qPCR) revealed that cerebral organoids from AD patients exhibited higher mRNA

levels of mature neuronal markers including *MAP2*, *CTIP2*, and *SATB2*, but not *GFAP*, an astrocyte marker (Supplementary Fig. 3A–D). Synaptophysin and PSD95 were also upregulated in AD organoid groups by week 4 (Supplementary Fig. 3E–G). While synaptophysin levels were increased in AD-E4 organoids (Supplementary Fig. 3F), *APOE4* did not influence PSD95 levels (Supplementary Fig. 3G). Together, these results suggest that organoid maturation and synaptic formations are accelerated in cerebral organoids from AD patients at the early stage.

**Increased Aβ amounts in cerebral organoids from AD-iPSCs.** To investigate the impacts of *APOE4* and disease status on Aβ accumulation and deposition, iPSC-derived cerebral organoids

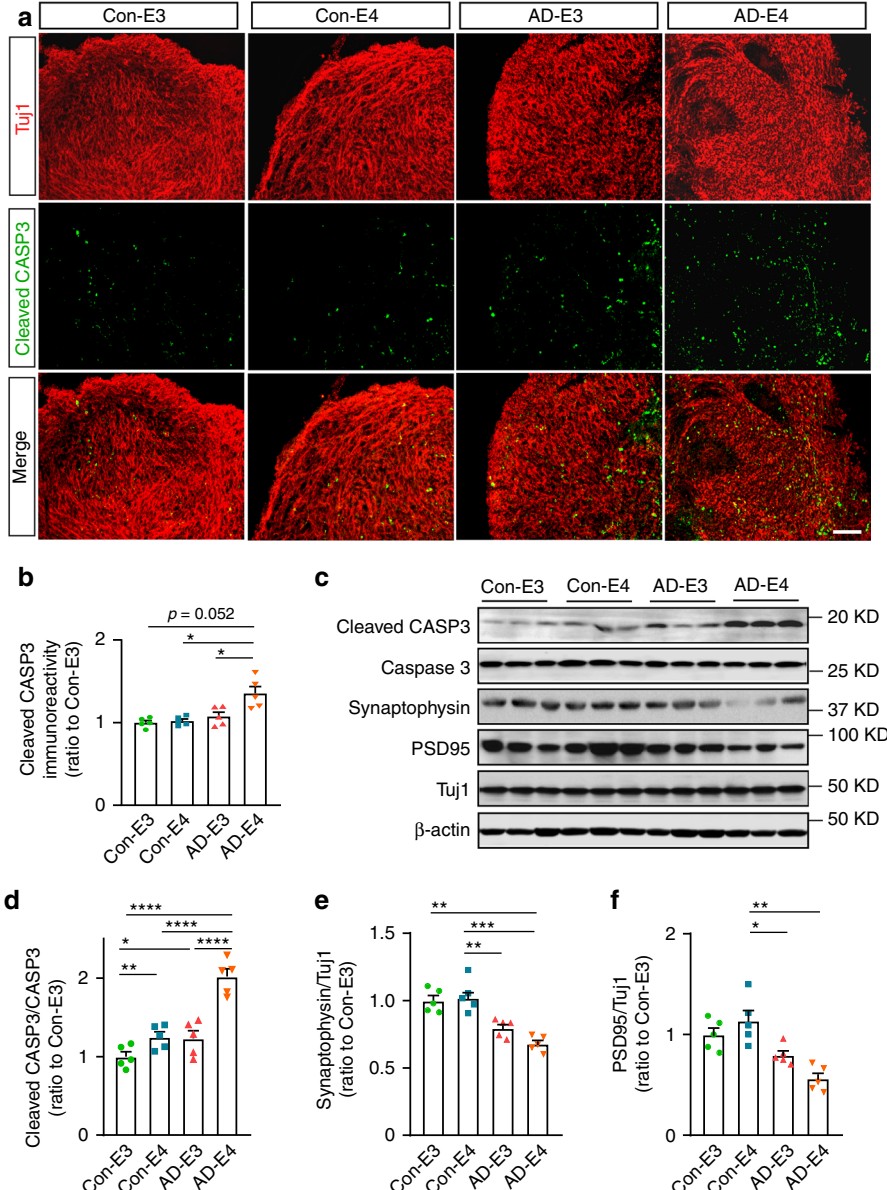

**Fig. 2 *APOE4* enhances apoptosis and synaptic loss in cerebral organoids from AD patients.** cerebral organoids were subjected to immunostaining and western blotting at week 12. **a** Representative images and quantification of cellular apoptosis evaluated by immunostaining of cleaved CASP3. Scale bar: 100 μm. **b** Cleaved CASP3 immunoreactivities were quantified from 5 cerebral organoids per line, and the averaged values were compared among the groups (*APOE4*: $p = 0.032$, AD: $p = 0.0569$, *APOE4* x AD: $p = 0.018$, Con-E3 vs. AD-E4: $p = 0.0523$, Con-E4 vs. AD-E4: $p = 0.0112$, AD-E3 vs. AD-E4: $p = 0.014$). All data are expressed as mean ± SEM ($N = 5$). **c–f** Cleaved CASP3, CASP3, synaptophysin, PSD95, and Tuj1 levels in the lysates of 4–5 cerebral organoids per line were analyzed by western blotting and quantified. All data are expressed as mean ± SEM ($N = 5$). **d** Cleaved CASP3 levels were normalized to total CASP3 levels and compared among groups (*APOE4*: $p < 0.0001$, AD: $p < 0.0001$, *APOE4* x AD: $p = 0.0020$, Con-E3 vs. Con-E4: $p = 0.009$, Con-E3 vs. AD-E3: $p = 0.0206$, Con-E4 vs. AD-E4: $p < 0.0001$, AD-E3 vs. AD-E4: $p < 0.0001$). Synaptophysin and PSD95 levels were normalized to Tuj1 levels and compared among groups (**e** *APOE4*: $p = 0.5841$, AD: $p = 0.0002$, *APOE4* x AD: $p = 0.0453$, Con-E3 vs. AD-E4: $p = 0.0069$, Con-E4 vs. AD-E4: $p = 0.0002$, Con-E4 vs. AD-E3: $p = 0.0077$. **f** *APOE4*: $p = 0.8794$, AD: $p = 0.0025$, *APOE4* x AD: $p = 0.0551$, Con-E4 vs. AD-E3: $p = 0.0404$, Con-E4 vs. AD-E4: $0.0019$). ANCOVA for *APOE4*, AD status, and *APOE4* x AD status was performed by including sex, sampling age, and source of iPSCs as co-variables, which was followed by two-sided Tukey–Kramer tests to compare between the groups with two factors (*APOE4* and AD status). *$p < 0.05$, **$p < 0.01$, ***$p < 0.001$, ****$p < 0.0001$.

were utilized for the analyses at different time points (weeks 4, 8 and 12). Organoids were sequentially lysed in RIPA buffer and formic acid (FA), and subjected to the measurements for Aβ40 and Aβ42. Using ELISA, we detected Aβ at an earlier time point than that reported by Lin et al.[28], in which they assessed Aβ accumulation by western blotting. Higher levels of Aβ40 at week 8 (Supplementary Fig. 4A) and Aβ42 at weeks 4 and 8 (Supplementary Fig. 4B) were observed in the RIPA-soluble fraction from

AD organoids as compared to those from healthy subject-derived cerebral organoids. The increases of Aβ40 (Fig. 3a), Aβ42 (Fig. 3b), and Aβ42/Aβ40 ratio (Fig. 3c) in AD organoid groups were magnified at week 12 independent of *APOE4*. Aβ40 and Aβ42 in detergent-insoluble FA fraction were undetectable at any time points. In addition, neither *APOE4* nor disease status affected APP derivatives including sAPPα, sAPPβ, and CTF-β at week 12 (Fig. 3d–f), whereas *APOE4* but not AD status was

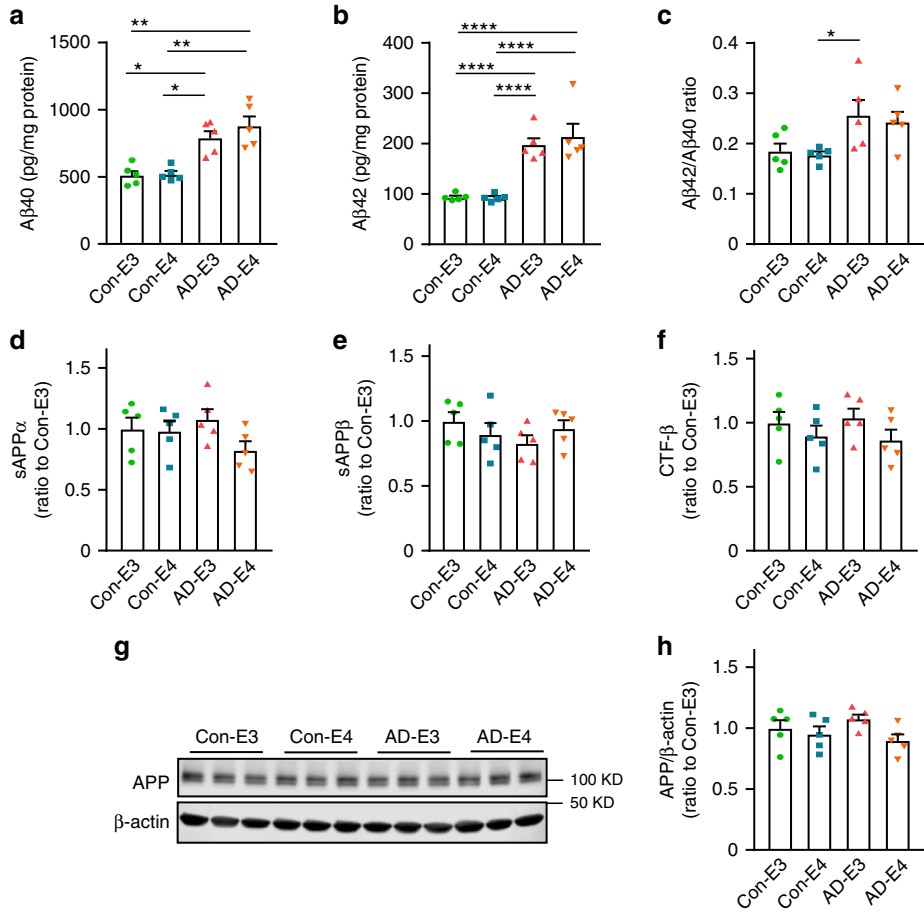

**Fig. 3 Increased Aβ accumulation in iPSC-derived cerebral organoids from AD patients.** Lysates of 4–5 cerebral organoids per iPSC line were analyzed by ELISA and western blotting at week 12. **a–c** Amounts of Aβ40 (**a**; *APOE4*: $p = 0.2167$, AD: $p = 0.0002$, *APOE4* x AD: $p = 0.4849$, Con-E3 vs. AD-E3: $p = 0.0287$, Con-E3 vs. AD-E4: $p = 0.0028$, Con-E4 vs. AD-E3: $p = 0.0171$, Con-E4 vs. AD-E4: $p = 0.0011$) and Aβ42 (**b**; *APOE4*: $p = 0.1014$, AD: $p < 0.0001$, *APOE4* x AD: $p = 0.7778$, Con-E3 vs. AD-E3: $p < 0.0001$, Con-E3 vs. AD-E4: $p < 0.0001$, Con-E4 vs. AD-E3: $p < 0.0001$, Con-E4 vs. AD-E4: $p < 0.0001$) in the RIPA fraction were measured by ELISA. Data were normalized to the total protein concentration of the respective sample. The ratio of Aβ42/Aβ40 was calculated accordingly (**c**; *APOE4*: $p = 0.9549$, AD: $p = 0.0034$, *APOE4* x AD: $p = 0.5331$, Con-E4 vs. AD-E3: $p = 0.0397$). **d–f** Amounts of sAPPα (**d**; *APOE4*: $p = 0.1081$, AD: $p = 0.8009$, *APOE4* x AD: $p = 0.1351$), sAPPβ (E; *APOE4*: $p = 0.7772$, AD: $p = 0.4410$, *APOE4* x AD: $p = 0.0936$) and CTF-β (**f**; *APOE4*: $p = 0.1150$, AD: $p = 0.5326$, *APOE4* x AD: $p = 0.9877$) in RIPA were measured by ELISA. Data are shown as ratios to Con-E3 after normalization to total protein concentration. **g, h** Amounts of full-length APP were assessed by western blotting analysis using 22C11 monoclonal antibody in RIPA lysate (*APOE4*: $p = 0.0331$, AD: $p = 0.6706$, *APOE4* x AD: $p = 0.3170$). All data are expressed as mean ± SEM ($N = 5$). ANCOVA for *APOE4*, AD status, and *APOE4* x AD status was performed by including sex, sampling age, and source of iPSCs as co-variables, which was followed by two-sided Tukey–Kramer tests to compare between the groups with two factors (*APOE4* and AD status). $*p < 0.05$, $**p < 0.01$, $****p < 0.0001$.

associated with lower levels of full-length APP (Fig. 3g, h). These results suggest that Aβ clearance mechanism rather than APP processing is altered in cerebral organoids from AD patients regardless of *APOE4* status, resulting in enhanced Aβ accumulation and increased Aβ42/Aβ40 ratio.

**Exacerbated tauopathy in cerebral organoids from *APOE4*-iPSCs.** To examine the tau pathology in the healthy subject-derived and AD patient-derived cerebral organoids carrying different *APOE* genotypes, we stained cerebral organoids with anti-phosphorylated tau (p-tau) AT8 at week 12 and found more p-tau accumulation in organoids with *APOE4* (Fig. 4a). Consistent with the results, western blotting with AT8 revealed that *APOE4* and AD status were correlated with higher p-tau/tau ratio in the organoids (Fig. 4b–d). When RIPA lysates (Fig. 4e) and FA fractions (Fig. 4f) were subjected to ELISA, *APOE4*, and AD status were independently associated with p-tau upregulation in the organoids at week 12. While RIPA-soluble p-tau increased in all groups in time, *APOE4* or AD patient-derived organoids

showed higher p-tau levels as early as week 4 (Supplementary Fig. 4C). Together, these results indicate that *APOE4* and AD status aggravate tau pathology in iPSC-derived organoids without interactive effects.

When apoE levels were measured in the organoids by ELISA, higher soluble apoE levels in RIPA fraction was associated with *APOE4* at week 12 (Fig. 5a); however, AD status rather than *APOE4* was associated with increased insoluble apoE in the FA fraction at week 12 (Fig. 5b). Both *APOE4* and AD status were associated with the increased soluble apoE levels at week 4 (Supplementary Fig. 4D). No significant differences in GFAP/Tuj1 ratio were observed among the four groups of cerebral organoids by immunostaining (Supplementary Fig. 5A–B), RT-qPCR (Supplementary Fig. 5C), and western blotting (Supplementary Fig. 5D), suggesting that *APOE4* or AD status leads to the higher apoE level without increasing astrocyte population in the cerebral organoids. Among all groups, significant positive correlations were observed between p-tau and apoE levels in RIPA (Fig. 5c) and FA fractions (Fig. 5d). Although there were no

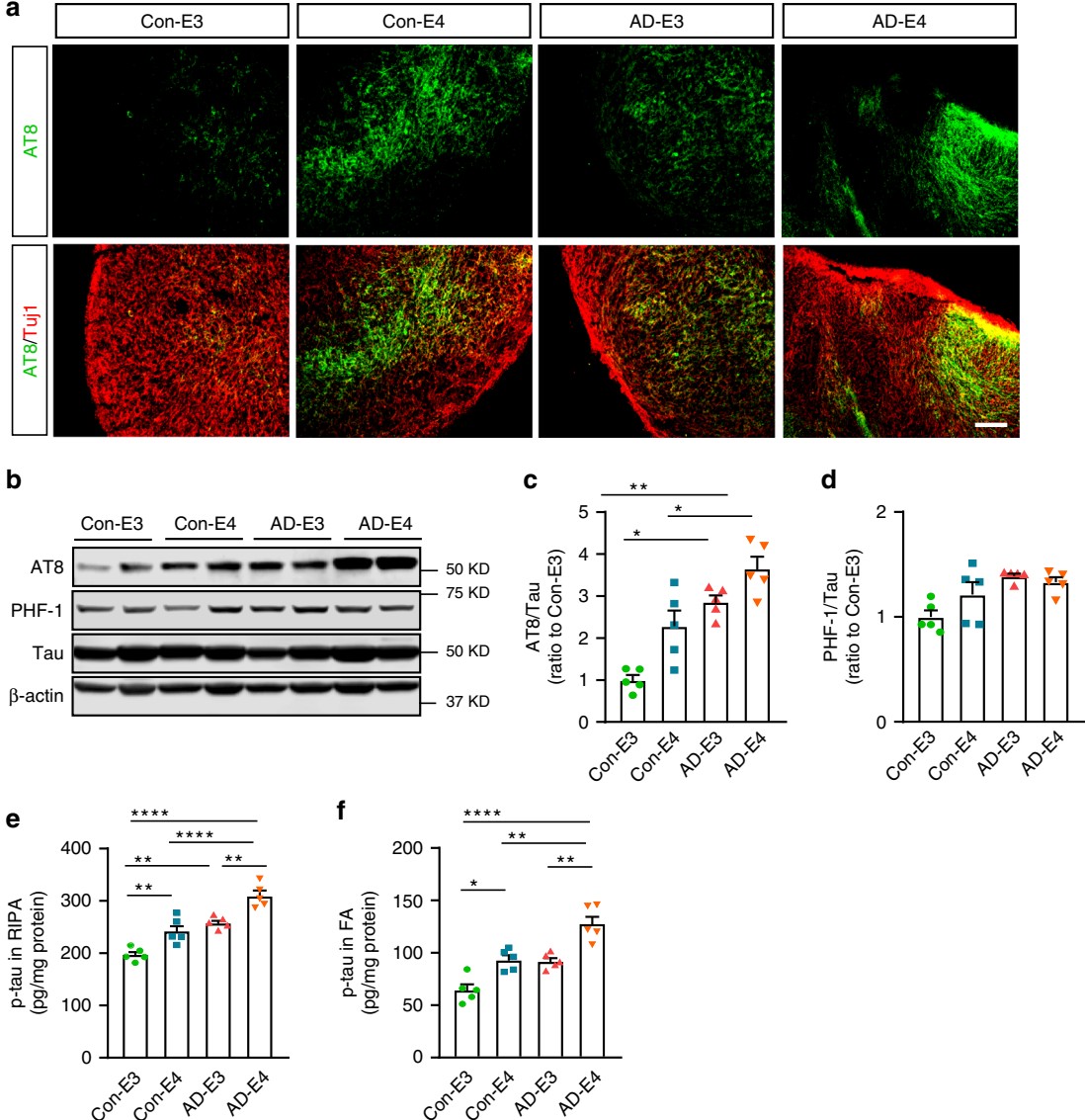

**Fig. 4 *APOE4* and AD status enhance p-tau levels in iPSC-derived cerebral organoids.** Cerebral organoids were subjected to analyses by immunostaining, western blotting, and ELISA at week 12. **a** Representative images of the immunostaining for p-tau with AT8 (Ser202/Thr205) antibody. Scale bar: 100 μm. **b–d** Total tau and p-tau levels in the RIPA lysates of 4–5 cerebral organoids per line were analyzed by western blotting and quantified. The p-tau levels detected by AT8 antibody (**c**; *APOE4*: $p = 0.0037$, AD: $p = 0.0010$, *APOE4* x AD $p = 0.6128$, Con-E3 vs. AD-E3: $p = 0.0290$, Con-E3 vs. AD-E4: $p = 0.0010$, Con-E4 vs. AD-E4: $p = 0.0273$) and PHF1 (Ser396/Ser404) (**d**; *APOE4*: $p = 0.3742$, AD: $p = 0.0404$, *APOE4* x AD: $p = 0.1823$) were normalized to total tau. **e, f** Amounts of p-tau in the RIPA fraction (**e**; *APOE4*: $p < 0.0001$, AD: $p < 0.0001$, *APOE4* x AD: $p = 0.7063$, Con-E3 vs. Con-E4: $p = 0.0029$, Con-E3 vs. AD-E3: $p = 0.0015$, Con-E3 vs. AD-E4: $p < 0.0001$, Con-E4 vs. AD-E4: $p < 0.0001$, AD-E3 vs. AD-E4: $p = 0.0010$) and the FA fraction (**f**; *APOE4*: $p < 0.0001$, AD: $p = 0.0005$, *APOE4* x AD: $p = 0.4038$, Con-E3 vs. Con-E4: $p = 0.0121$, Con-E3 vs. AD-E4: $p < 0.0001$, Con-E4 vs. AD-E4: $p = 0.0024$, AD-E3 vs. AD-E4: $p = 0.0012$) from 4 to 5 cerebral organoids per line were measured by ELISA. Data were normalized to individual total protein concentration. All data are expressed as mean ± SEM ($N = 5$). ANCOVA for *APOE4*, AD status, and *APOE4* x AD status was performed by including sex, sampling age, and source of iPSCs as co-variables, which was followed by two-sided Tukey–Kramer tests to compare between the groups with two factors (*APOE4* and AD status). *$p < 0.05$, **$p < 0.01$, ****$p < 0.0001$.

evident correlations between Aβ and apoE levels (Fig. 5e, f), Aβ levels (Fig. 5g, h) were also positively correlated with p-tau levels in RIPA fraction. These results suggest that Aβ and apoE might associate with p-tau through different molecular pathways.

**Altered transcriptomes in cerebral organoids from AD-iPSCs.** To further address the impact of *APOE4* and/or AD status on transcriptional profiles in iPSC-derived organoids, we performed RNA-sequencing (RNA-seq) at week 12. We identified 1331 and 717 differentially expressed genes (DEGs) between Con-E3 and Con-E4, and between AD-E3 and AD-E4, respectively, with an

overlap of 302 genes (Supplementary Data 1, Supplementary Fig. 5A). Among the overlapped genes, 265 genes changed in the same direction. When comparing between healthy subject-derived and AD patient-derived cerebral organoids, we identified 1188 DEGs between Con-E3 and AD-E3, and 501 DEGs between Con-E4 and AD-E4, with an overlap of 317 genes 304 of which were changed in the same direction (Supplementary Data 1, Supplementary Fig. 6B). Furthermore, pathway analysis revealed "Amyloid proteins" as the top-ranked network affected by the disease status and *APOE4* interaction (Supplementary Fig. 6C). The top-ranked pathways enriched by DEGs were

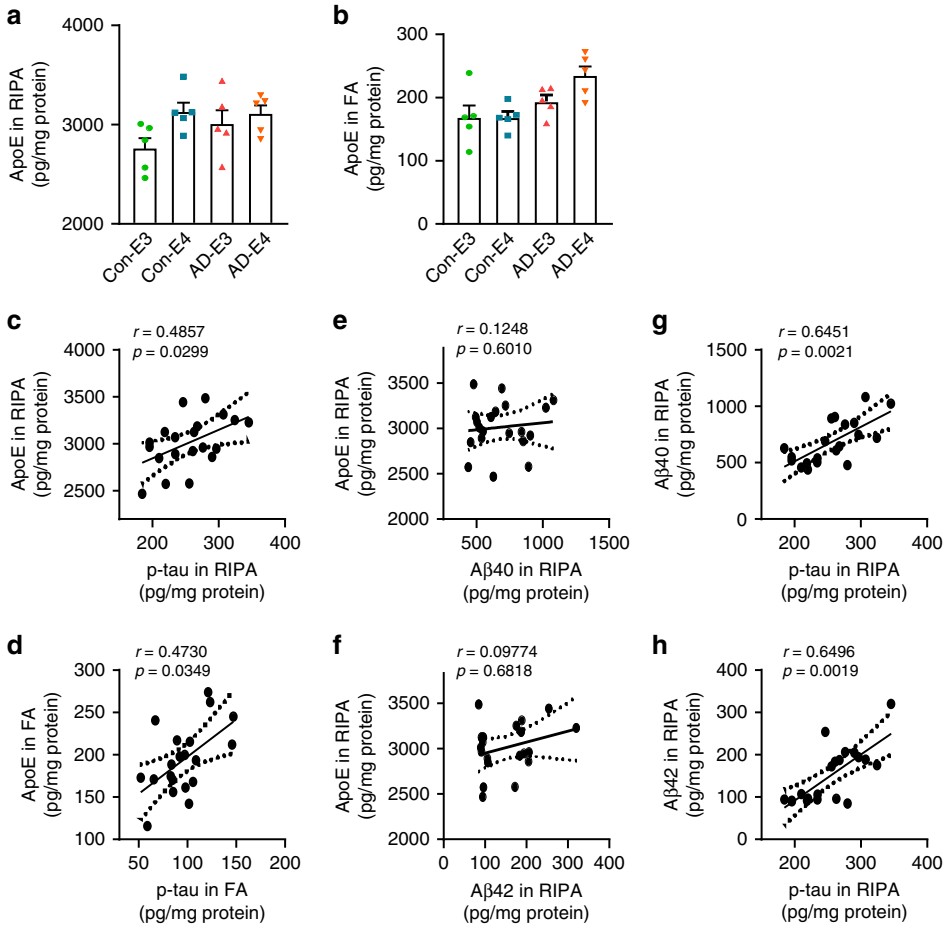

**Fig. 5 Positive correlation between apoE and p-tau levels in the iPSC-derived cerebral organoids. a, b** Amounts of apoE in the RIPA fraction (**a**; *APOE4*: *p* = 0.0322, AD: *p* = 0.3691, *APOE4* x AD: *p* = 0.4301) and the FA fraction (**b**; *APOE4*: *p* = 0.641, AD: *p* = 0.0244, *APOE4* x AD: *p* = 0.4023) of 4–5 cerebral organoids per line were measured by ELISA at week 12. Data were normalized to individual total protein concentration. All data are expressed as mean ± SEM (*N* = 5). ANCOVA for *APOE4*, AD status, and *APOE4* x AD status was performed by including sex, sampling age, and source of iPSCs as co-variables, which was followed by two-sided Tukey–Kramer tests to compare between the groups with two factors (*APOE4* and AD status). **c–h** Two-sided Spearman correlation analysis (before adjustment) of apoE vs. p-tau in RIPA (**c**) and FA (**d**) fractions, apoE vs. Aβ40 (**e**) or Aβ42 (**f**) in RIPA fraction, and p-tau vs. Aβ40 (**g**) or Aβ42 (**h**) in RIPA fraction.

"Actin filaments" in Con-E3 vs. Con-E4 comparison (Supplementary Fig. 6D), "Regulation of epithelial-to-mesenchymal transition" in AD-E3 vs. AD-E4 comparison (Fig. S6E), "Axonal guidance" in Con-E3 vs. AD-E3 comparison (Supplementary Fig. 6F), and "Endoplasmic reticulum stress pathway" in Con-E4 vs. AD-E4 comparison (Supplementary Fig. 6G). "Innate inflammatory response" also showed up in all of the top 10-ranked pathways, suggesting a potential involvement of inflammation in *APOE4* effects and AD pathogenesis (Supplementary Fig. 6D–G). The expression of selected most significant changed genes (*DND1*, *NR2F6*, *VPS9D1*, *CTH*, and *CDHR3*) was further validated by RT-qPCR, the results of which were consistent with those from RNA-seq (Supplementary Fig. 6H–L).

To assess potential contributions of different brain cell types for these DEGs, we analyzed the RNA-seq data using CIBER-SORT and CellCODE program. We found that genes detected in the cerebral organoids were mainly derived from neurons (median: 0.9231) and astrocytes (median: 0.06698), whereas oligodendrocytes compose negligible proportion of the cell population (Supplementary Fig. 7A). No significant differences in neuron and astrocyte proportion were observed among different groups (Supplementary Fig. 7B–C). As more DEGs were designated to neurons than those assigned to astrocytes in all comparisons (Supplementary Fig. 7D–G), those results

suggest that neurons predominantly contributed to the phenotypes of cerebral organoids from different groups compared to astrocytes.

Weighted Gene Co-expression Network Analysis (WGCNA) identified two gene modules that were significantly associated with disease status: module magenta was positively correlated with AD, while module yellow was positively correlated with healthy subject-derived cerebral organoids (negatively correlated with AD) (Fig. 6a). Genes in the yellow module were enriched for RNA metabolism dysregulation (Fig. 6b), which included *ERCC4*, *DGCR8*, *POLR3A*, *CLP1*, *HSPA4*, *PNO1*, *VPS18*, *RAD17*, *LCMT2*, and *RPUSD2* as top-ranked hub genes (Fig. 6c). We further validated expressions of selected genes by RT-qPCR and western blotting, and confirmed decreased mRNA and protein levels in AD-E4 organoids (Fig. 6d, f, j–l), Genes in the magenta module were enriched for those related to DNA and nucleosome metabolism pathways (Supplementary Fig. 8A). *CHIC1*, *H2BC7*, *H2BC8*, *ASB3*, *H2BC18*, H2BC21, H1-3, *ARRDC3*, and *TBL1XR1-AS1* were identified as top-ranked hub genes (Supplementary Fig. 8B). We further confirmed the increased mRNA levels of *CHIC1*, *ASB3*, and *ARRDC3* by RT-qPCR in AD organoids, in particular those with *APOE4* (Supplementary Fig. 8C–E). Since some of the hub genes in the yellow module (*PNO1*, *DGCR8, VPS18,* and *HSPA4*) are

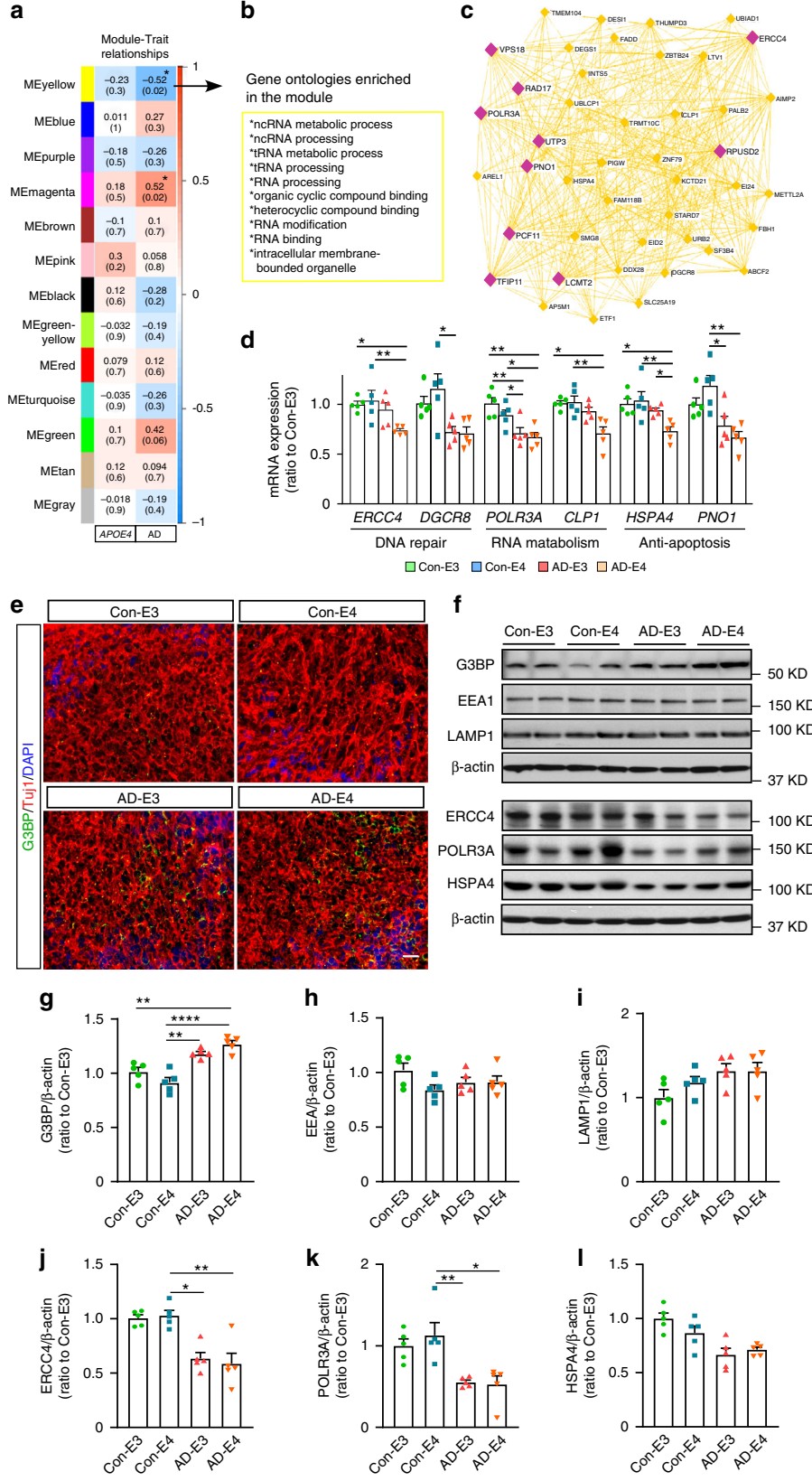

closely related to lysosomal stress granules formation under cellular stress conditions[29,30], we immunostained the cerebral organoids for a stress granule marker Ras GTPase-activating protein-binding protein 1 (G3BP) in the cerebral organoids at week 12, and found an increase of G3BP-positive punctates in AD organoids (Fig. 6e). We further quantified G3BP, endosome marker protein early endosome antigen 1 (EEA1), and lysosome marker protein lysosomal-associated membrane protein 1 (LAMP1) by western blotting, and found the interactive effect between *APOE4* and AD status; G3BP

**Fig. 6 Transcriptomics profiling of the iPSC-derived cerebral organoids by WGCNA. a** Module-trait relationships between modules and *APOE* genotype, and between modules and AD status are shown. **b**, **c** Top gene ontologies and interaction of genes enriched in the yellow module genes. Purple nodes are hub genes (top 10 highest connectivity). **d** The mRNA expressions of *ERCC4* (*APOE4*: $p = 0.0576$, AD: $p = 0.0064$, *APOE4* x AD: $p = 0.0742$, Con-E3 vs. AD-E4: $p = 0.0164$, Con-E4 vs. AD-E4: $p = 0.0049$), *DGCR8* (*APOE4*: $p = 0.1088$, AD: $p = 0.0469$, *APOE4* x AD: $p = 0.3298$, Con-E4 vs. AD-E3: $p = 0.0409$), *POLR3A* (*APOE4*: $p = 0.3926$, AD: $p = 0.0004$, *APOE4* x AD: $p = 0.1349$, Con-E3 vs. AD-E3: $p = 0.0048$, Con-E3 vs. AD-E4: $p = 0.0084$, Con-E4: AD-E3: $p = 0.0221$, Con-E4 vs. AD-E4: $p = 0.0484$), *CLP1* (*APOE4*: $p = 0.0953$, AD: $p = 0.0039$, *APOE4* x AD: $p = 0.1831$, Con-E3 vs. AD-E4: $p = 0.0157$, Con-E4 vs. AD-E4: $p = 0.0066$), *HSPA4* (*APOE4*: $p = 0.0501$, AD: $p = 0.0077$, *APOE4* x AD: $p = 0.0410$, Con-E3 vs. AD-E4: $p = 0.0171$, Con-E4 vs. AD-E4: $p = 0.0038$, AD-E3 vs. AD-E4: $p = 0.0339$), and *PNO1* (*APOE4*: $p = 0.8087$, AD: $p = 0.0022$, *APOE4* x AD: $p = 0.1706$, Con-E4 vs. AD-E3: $p = 0.0218$, Con-E4 vs. AD-E4: $p = 0.0038$) were quantified by RT-qPCR. **e** Representative images of the immunostaining for G3BP and Tuj1. Scale bar: 50 μm. **f–i** G3BP, EEA1, and LAMP1 levels were analyzed by western blotting. The levels of G3BP (**g**; *APOE4*: $p = 0.6345$, AD: $p < 0.0001$, *APOE4* x AD: $p = 0.0133$, Con-E3 vs. AD-E4: $p = 0.0025$, Con-E4 vs. AD-E3: $p = 0.0019$, Con-E4 vs. AD-E4: $p < 0.0001$), EEA1 (**h**; *APOE4*: $p = 0.2792$, AD: $p = 0.6789$, *APOE4* x AD: $p = 0.0671$), and LAMP1 (**i**; *APOE4*: $p = 0.2558$, AD: $p = 0.0954$, *APOE4* x AD: $p = 0.5646$) were normalized to β-actin. **j–l** ERCC4, POLR3A, and HSPA4 were analyzed by western blotting. The levels of ERCC4 (**j**; *APOE4*: $p = 0.9762$, AD: $p = 0.0016$, APOE4 x AD: $p = 0.3378$, Con-E4 vs. AD-E3: $p = 0.0241$, Con-E4 vs. AD-E4: $p = 0.0054$), POLR3A (**k**; *APOE4*: $p = 0.2095$, AD: $p = 0.0018$, APOE4 x AD: $p = 0.7764$, Con-E4 vs. AD-E3: $p = 0.0048$, Con-E4 vs. AD-E4: $p = 0.0149$), and HSPA4 (**l**; *APOE4*: $p = 0.6526$, AD: $p = 0.0152$, APOE4 x AD: $p = 0.2371$) were normalized to β-actin. All data are expressed as mean ± SEM ($N = 5$). ANCOVA for *APOE4*, AD status, and *APOE4* x AD status was performed by including sex, sampling age, and source of iPSCs as co-variables, which was followed by two-sided Tukey–Kramer tests to compare between the groups with two factors (*APOE4* and AD status). *$p < 0.05$, **$p < 0.01$, ****$p < 0.0001$.

significantly increased in organoids from AD patients carrying *APOE4* (Fig. 6f, g). No significant changes were observed in EEA1 (Fig. 6h) and LAMP1 levels (Fig. 6i). These results imply that the disruption of RNA metabolism in cerebral organoids from AD patients accelerates stress granule formation especially in the presence of *APOE4*.

**Conversion of *APOE4* to *APOE3* attenuates AD pathologies.** Using a parental iPSC line with *APOE ε4/ε4* from a sporadic AD patient (Par-E4/4) and the isogenic iPSC line with *APOE ε3/ε3* (Iso-E3/3)[19], we tested whether converting *APOE4* to *APOE3* could ameliorate *APOE4*-related phenotypes in iPSC-derived organoids. While no obvious size difference was observed between cerebral organoids from the parental line and isogenic line at week 12 (Fig. 7a), immunostaining showed the reduction of cleaved CASP3 immunoreactivity in the edge of cerebral organoids from Iso-E3/3 compared to Par-E4/4 organoids (Fig. 7b, Supplementary Fig. 10C). The levels of cleaved CASP3 in the center of the cerebral organoids at week 12 were measured to evaluate the possible influence of necrotic core. No significant differences in the center of cerebral organoids from the parental line and isogenic line were observed (Supplementary Fig. 10B). Correlation analysis showed no significant correlation between the levels of cleaved CASP3 at the center with those at the edge, suggesting the apoptosis at the edge of cerebral organoids is not driven by the necrotic core at the center (Supplementary Fig. 10D). Western blotting also showed a lower cleaved CASP3/CASP3 ratio in Iso-E3/3 organoids than Par-E4/4 organoids (Fig. 7c). In addition, Iso-E3/3 organoids showed the decreased Aβ40 (Fig. 7d) and Aβ42 (Fig. 7e) levels in the RIPA lysate, although the Aβ42/40 ratio (Fig. 7f) remained unaffected by *APOE4* conversion. Whereas RIPA-soluble apoE levels were not affected (Fig. 7g), insoluble apoE in FA fraction was decreased in the Iso-E3/3 organoids (Fig. 7h). Immunostaining (Fig. 7i) and western blotting (Fig. 7j) showed that *APOE4* conversion to *APOE3* reduced p-tau levels and p-tau/tau ratios in both RIPA (Fig. 7k) and FA fractions (Fig. 7l) in Iso-E3/3 organoids compared with Par-E4/4 organoids, which were confirmed by ELISA. Converting *APOE4* to *APOE3* reduced the levels of G3BP in cerebral organoids when analyzed by western blotting (Fig. 7m). Taken together, these results indicate that *APOE4*-related phenotypes observed in AD patient-derived organoids can largely be reversed through genome editing to *APOE3*.

## Discussion

While AD is neuropathologically diagnosed through postmortem assessments in patients with dementia[31], recent research framework has established antemortem AD classification using biomarkers for Aβ deposition (A), pathologic tau (T), and neurodegeneration (N)[32]. Thus, to explore the complex AD pathogenesis and develop therapeutic interventions for the disease, the establishment of human models system recapitulating ATN phenotypes has become essential. Toward this, we have comprehensively investigated AD-related pathogenic pathways using iPSC-derived 3-D cerebral organoids from sporadic AD patients with or without *APOE4*. cerebral organoids not only simulate intrinsic spatial patterning, but also display acquisition of cell identity in a timed manner that closely mimics the temporal patterning with sequential neuronal layer formation, accompanied with matured astrocytes[21,22,33]. Furthermore, the phenotypes related to aberrant extracellular protein aggregation can be recapitulated only in 3-D culture systems[28,34,35]. Using iPSC-derived cerebral organoid system, we identified multiple *APOE4*- and/or AD disease status-dependent pathogenic pathways, revealing complex etiology associated with *APOE4*, the strongest genetic risk factor for late-onset AD.

Consistent with results from 2-D cultures of iPSC-derived neurons[13,18,28,36], we found the elevated Aβ40 and Aβ42 levels in RIPA-soluble fraction in AD patient-derived organoids regardless of *APOE4* status. Since APP processing was not altered, AD brain organoids may have compromised machineries for enzymatic Aβ degradation and/or cellular Aβ clearance. Of note, a significant reduction of soluble Aβ levels was observed in the cerebral organoids from a sporadic AD patient upon isogenic conversion of *APOE4* to *APOE3*. Thus, it is possible that *APOE4* and other gene variants synergistically facilitate Aβ accumulation in AD organoids. Notably, insoluble Aβ and amyloid plaque pathology was undetectable by ELISA and immunostaining in our organoid models from sporadic AD patients at week 12. While Aβ plaque deposition is detected 2–4 months after the differentiation in iPSC-organoid models from familial AD and Down syndrome patients[35,37], those from sporadic AD patients likely have Aβ deposition 6 months after differentiation[28]. Thus, longer differentiation duration may be required to assess amyloid deposition in iPSC-derived organoids from sporadic AD cases.

Another pathological hallmark of AD is the abnormal phosphorylation, mislocalization, and aggregation of tau[38,39], although tauopathy is also detected in non-AD neurodegenerative diseases including frontotemporal dementia (FTD) and progressive

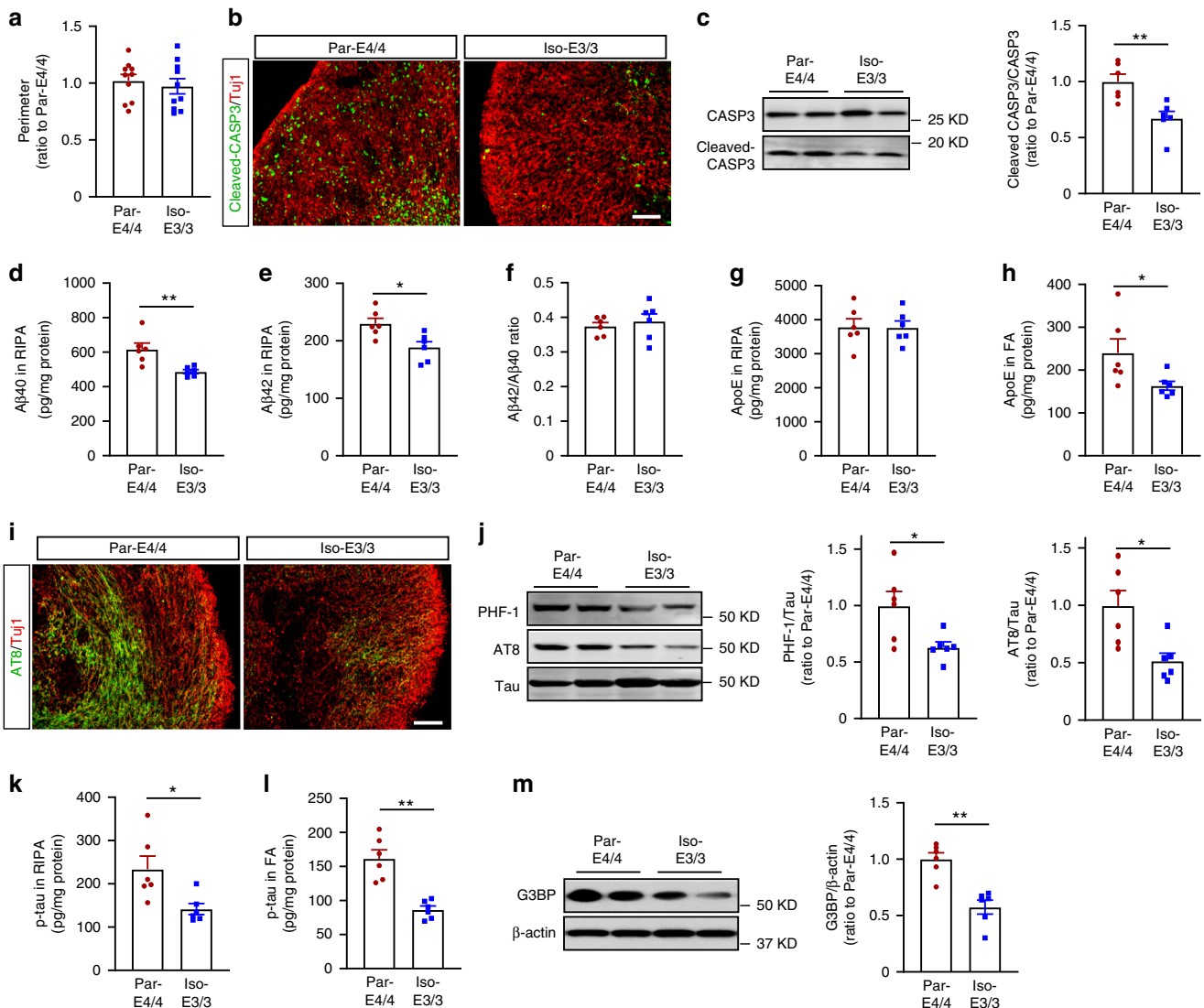

**Fig. 7 Isogenic conversion of *APOE4* to *APOE3* attenuates AD-related phenotypes in cerebral organoids.** The iPSC-derived cerebral organoids from an AD patient carrying *APOE ε4/ε4* (Par-E4/4) and the *APOE ε3/ε3* isogenic line (Iso-E3/3) were analyzed at week 12. **a** The perimeters of cerebral organoids were measured ($n = 10$). **b** Representative images of cellular apoptosis with the immunostaining for cleaved CASP3. Scale bar: 100 μm. **c** Cleaved CASP3 and CASP3 levels in the lysates were analyzed by western blotting and quantified ($p = 0.0043$). **d–f** Amounts of Aβ40 (**d**: $p = 0.0043$) and Aβ42 (**e**: $p = 0.0152$) in RIPA fraction were measured by ELISA. Data were normalized to individual protein concentrations. The ratio of Aβ42/Aβ40 (**f**) was calculated accordingly. **g, h** Amounts of apoE in RIPA (**g**) and FA (**h**: $p = 0.0411$) were measured by ELISA. Data were normalized to individual total protein concentrations. **i** Representative images of the immunostaining for p-tau with AT8 antibody. Scale bar: 100 μm. **j** Total tau and p-tau in RIPA lysates were analyzed by western blotting and quantified. The p-tau detected by AT8 antibody and PHF1 antibody were normalized to total tau (AT8: $p = 0.0152$, PHF1: $p = 0.0411$). **k, l** Amounts of p-tau in RIPA (**k**: $p = 0.0152$) and FA (**l**: $p = 0.0022$) were measured by ELISA. Data were normalized to individual total protein concentration. **m** G3BP, levels in RIPA lysates were analyzed by western blotting and quantified. The levels of G3BP were normalized to β-actin ($p = 0.0022$). **c–h**, **j–m** Lysates of 4–5 cerebral organoids were analyzed as one sample. All data are expressed as mean ± SEM ($n = 6$). Two-sided Mann–Whitney U tests were performed to determined statistical significance, *$p < 0.05$, **$p < 0.01$.

supranuclear palsy (PSP)[40]. Indeed, tau phosphorylation is accelerated in iPSC-derived neurons from AD patients[13,41]. Furthermore, *APOE4* has been shown to exacerbate AD-related tau pathologies in cell and mouse models as well as human patients[10,42]. Our results also found that AD status and *APOE4* are independently associated with the increased levels of p-tau in iPSC-derived cerebral organoids. Given that neither *APOE4* nor apoE amounts influenced Aβ levels in the iPSC-derived cerebral organoids, *APOE4* may accelerate the p-tau accumulation in an Aβ-independent manner. Of note, apoE and p-tau levels show positive correlations in both RIPA and FA fractions, which is consistent with the results from mouse studies showing that *Apoe*

deficiency ameliorates tauopathy[42]. In addition, Wang et al. has shown that p-tau levels were increased in *APOE*-deficient iPSC-derived neurons when treated with lysates of iPSC-derived neurons with *APOE4*[19]. Thus, although further studies are necessary, *APOE4* likely induces a gain-of-toxic effect on p-tau accumulation in iPSC-derived cerebral organoids.

Intriguingly, we found marked increase in cellular apoptosis and reductions of synaptic proteins in mature organoids from AD patients compared to healthy subject-derived cerebral organoids. In supporting our findings, iPSC-derived neurons from a sporadic AD patient have been reported for their hyper-excitable calcium signaling phenotype, increased cytotoxicity and apoptosis, and

reduced neurite length in comparison to those from healthy individuals[36]. Wang et al. also found expression of apoE4 was associated with increased GABAergic neuron degeneration[19]. More supportively, the gain-of-toxic-function of apoE4 could be ameliorated when apoE4 was converted to apoE3 by gene editing, which is highly consistent with what we found in the cerebral organoid model[19]. In contrast, at the early neuronal differentiation stage, increased mature neuronal markers and synaptic proteins were detected in AD-derived organoids. Given that similar observations were reported in iPSC-derived neurons from AD subjects[43], specific factors facilitating excess neuronal maturation in the early stage may be commonly preserved in iPSCs from AD patients, resulting in enhanced neurodegeneration in the late stage. While *APOE4* likely leads to early maturation of neurons[28] partially consistent with our results, we demonstrated that *APOE4* further exacerbates the effects of AD status on apoptosis induction and synaptic loss in cerebral organoids at week 12. Thus, it is tempting to speculate that accelerated neuronal differentiation/maturation is synergistically induced by AD-related factors and *APOE4* in the pre-symptomatic stage, and that the mechanistic exhaustion followed by neuronal dysfunction may contribute to disease development, although further studies are needed.

Our transcriptomic analysis demonstrated that an enrichment of gene sets involving RNA metabolisms is predominantly affected by AD status in the iPSC-derived cerebral organoids; while *APOE4* did not have a strong effect on the profiles. RNA metabolism process is highly dynamic and requires a complex interplay among RNA-binding proteins (RBPs)[44]. Thus, the disruption of RBP homeostasis is frequently involved in the pathology and genetics of neurodegenerative diseases including FTD and amyotrophic lateral sclerosis (ALS), where aggregation-prone RBPs often co-localize with stress granules[44–46]. Interestingly, we also found a significant increase in stress granules in organoids from AD patients. While various RBPs have been shown to deposit in tau inclusions[47–49], internalized tau sensitizes cells to stress by promoting formation and stability of stress granules[49]. Thus, enhanced p-tau accumulation may trigger transcriptional changes and stress granule formation in organoids from AD patients. There were also reductions of mRNAs in genes related to DNA repair (*ERCC4* and *DGCR8*)[50,51] and anti-apoptosis (*PNO1* and *HSPA4*)[52,53] in the organoids from AD patients. However, further studies are necessary to determine if those are causatively or consequently involved in neurodegeneration.

Although our results demonstrate that iPSC-derived cerebral organoid is a useful modeling system to investigate AD- and *APOE4*-related phenotypes, we should state several limitations. First, the core of cerebral organoids shows necrosis-like changes after week 12, likely due to the lack of vascular systems maintaining proper supplies of nutrition and gas exchange as well as immune cells to eliminate cell debris and toxic molecules. To address this caveat, emerging technology has further developed cerebral organoid models in which vascular cells[19,54] or microglia are incorporated[28,55,56]. Thus, in future studies, we plan to utilize the iPSC-derived cerebral organoid system with neuroimmune and/or neurovascular system to establish a more advanced platform for AD research. Second, there was heterogeneity in organoid size and growth rate even though identical iPSC lines were used. We also noticed that maturation of astrocytes and the detection of Aβ or apoE in cerebral organoids may vary depending on differentiation protocols[28,33,57]. For example, Lin et al.[28] found lower soluble apoE levels in cerebral organoids with *APOE4* compared to those with *APOE3*, which is different from our results. The differences in the medium used for differentiation and the time points of medium change may potentially contribute to the different results under the stress conditions induced by *APOE4* or AD status. The technical differences and the intrinsic variability using iPSC-derived cerebral organoid models should be considered whenever comparing results from different studies. Third, because of the limited availability of patient-derived iPSC lines, we could not match sex among our study groups in our study. Nonetheless, we did adjust sex as a variable in our analysis. Further optimization and standardization in the iPSC-derived cerebral organoid modeling system should be able to overcome these weaknesses.

In conclusion, our study established a true 3-D human cerebral organoid system to address AD pathogenesis. We successfully recapitulated AD-related pathologies related to ATN classification in our model system. *APOE4* predominantly aggravates p-tau accumulation, while AD status is associated with higher levels of Aβ and p-tau, apoptosis, synaptic loss and increased stress granule formation. Importantly, *APOE4* synergistically accelerates apoptosis and stress granule formation with AD status in our 3D model. Thus, exploring molecular mechanisms in the interaction between *APOE4*- and AD-related pathways should provide clues as to how *APOE4* vastly increases AD risk. Because isogenic conversion of *APOE4* into *APOE3* reverses much of the AD-related phenotypes in cerebral organoids from AD, *APOE4* might be a promising therapeutic target for AD.

## Methods

**Generation of iPSCs from human skin fibroblasts.** Human skin biopsies from normal individuals and AD patients with *APOE* ε3/ε3 or ε4/ε4 genotype were obtained from Mayo Clinic patients under IRB protocols with patient consent for research, which was approved by the Mayo Clinic Institutional Review Board. *APOE* genotype was confirmed by Sanger sequencing using DNA samples from fibroblast lines. Cells were cultured in fibroblast medium containing 10% fetal bovine serum (FBS) (Gemini Bio-Products). The iPSCs were generated by electroporation of three episomal vectors into the fibroblasts using the NHDF nucleofector kit (Lonza)[23,58]. Three micrograms of expression plasmid mixtures were electroporated into $6 \times 10^5$ fibroblasts with 100 μl transfection reagents. After transfection, fibroblasts were plated onto a 100 mm Matrigel (Corning) coated dish. After 5 days of culture, the fibroblast medium was replaced with TeSR-E7 complete medium (Stemcell Technologies) and changed every day. IPSC colonies were isolated and expanded after 3–4 weeks in culture. The iPSC colonies were passaged using Dispase (Stemcell Technologies) and subjected to rock inhibitor Y27632 (Sigma-Aldrich) treatment for the first 24 h.

**Trilineage differentiation of human iPSCs.** Three germs layer differentiation was used to confirm the pluripotency of iPSCs using the STEMdiff Trilineage Differentiation kit (Stemcell Technologies) according to manufacturer's instructions with some modification[59]. When cells were ~70% confluent, iPSCs were passaged using Accutase (Stemcell Technologies) and plated into an AggreWell™800 plate (Stemcell Technologies) to form embryonic bodies (EB) in mTeSR1 medium for 2 days. EBs were washed out from the AggreWell plate and transferred to 6-well non-tissue culture treated plates (Corning) in specific differentiation mediums for each lineage. EBs were subjected to differentiation into mesoderm and endoderm lineages for 5 days or an ectoderm lineage for 7 days, then seeded onto Matrigel-coated plates for further analysis. Differentiation was assessed by immunostaining for germ layer-specific markers (Endoderm: SOX17; Mesoderm: Brachyury; Ectoderm: Nestin/Sox2).

**Cerebral organoid culture.** Cerebral organoids were generated according to manufacturer's instructions of the commercial STEMdiff™ Cerebral Organoid Kit (Stemcell Technologies). On day 0, any pre-differentiated cells in the hiPSCs culture were removed by scraping under a microscope. Human iPSC colonies were dissociated into single-cell suspension with Accutase. In total, 9000 cells were then plated into each well of a U-bottom ultra-low-attachment 96-well plate in EB formation media (medium A) supplemented with 10 μM Y27632. An extra 100 μl EB medium was added on day 2 and day 4, respectively. EBs were moved to 24-well low attachment plates in neural induction medium (medium B) for another 3–5 days. EBs were further embedded with 20 μl of matrigel and cultured in neural expansion medium (medium C + D) for 3 days in 6-well low attachment plates for organoid formation. In the final stage, the organoids were transferred to 10-cm dishes with neural culture medium (medium E), and moved to an orbital shaker for further culture. Medium E was replaced with neuronal maturation medium after 4 weeks, which was composed of DMEM/F12 + Neurobasal Medium (1:1) supplemented with N2, B27, BDNF (20 ng/ml), GDNF (20 ng/ml), ascorbic acid (200 μM), and dbcAMP (100 nM) (Sigma-Aldrich)[60]. Cerebral organoids were

collected at different time points for immunostaining and other biochemical analysis. We differentiated 20 lines of iPSCs into cerebral organoids within a close time frame, and used cerebral organoids with normal size range (typically 2–4 mm of diameter) for our experiments.

**Immunostaining**. At pre-defined time points, cerebral organoids were fixed in 4% paraformaldehyde for 30 min then washed with PBS three times. After fixation, organoids were dehydrated with 30% sucrose in PBS at 4 °C. Organoids were then embedded with optical cutting temperature (OCT) compound (VWR) and frozen on dry ice. Frozen tissue was sectioned at 30 µm using a cryostat and collected on ultra-frosted glass microscope slides. Sections were stored at −20 °C. For immunostaining, sections were permeabilized in 0.25% Triton X-100 and blocked with blocking buffer containing 4% normal donkey serum, 2% BSA, and 1 M glycine in PBS. Sections were then incubated with primary antibodies in blocking buffer overnight at 4 °C. The information of primary antibodies and their dilutions used in this study are as follows: Nanog (Cell Signaling, 4903, 1:300), TRA-1-60 (Abcam, ab16288, 1:300), SSEA4 (Abcam, ab16287, 1:300), Sox17 (Abcam, ab84990, 1:300), Brachyury (R&D, AF2085, 1:300), Nestin (Abcam, ab18102, 1:500), Sox2 (Abcam, ab97959, 1:500), TUJ1 (Abcam, ab78078, 1:1000), TUJ1 (Sigma, T2200, 1:1000), CTIP2 (Abcam, ab18465, 1:100), SATB2 (Abcam, ab34735, 1:100), GFAP (Millipore, MAB360, 1:300), cleaved Caspase-3 (Cell Signaling Technology, 9579, 1:300), AT8 (Thermo Fisher Scientific, MN1020, 1:300), and G3BP (BD, 611126, 1:300). After washing three times with PBS, samples were incubated with fluorescently conjugated secondary antibodies (Alexa Fluor 488 and 594 conjugates, Invitrogen, 1:500) for 2 h at room temperature and washed three times with PBS before mounting with the glass coverslip. Fluorescent signals were detected by fluorescence microscopy (model IX71 Invert, Olympus), confocal laser scanning fluorescent microscopy (model LSM510 Invert, Carl Zeiss), and Keyence fluorescence microscopy (model BZ-X, Keyence). The fluorescence signal was quantified using ImageJ software.

**Tissue processing**. Cerebral organoids were harvested at pre-defined time points and lysed with RIPA Lysis and Extraction Buffer supplemented with Protease and Phosphatase Inhibitor Cocktails for Cell Lysis (Roche). Lysed samples were sonicated and incubated for 60 min on ice. Samples were centrifuged in an ultracentrifuge (Beckman–Coulter) at $100,000 \times g$, for 1 h at 4 °C. Supernatants (soluble fraction) were collected and the pellet was resuspended in 50 µl of 70% formic acid (FA), sonicated, and neutralized with 2.5 M Tris buffer (pH 8.5) (insoluble fraction). Total protein concentration in the soluble fraction was determined using a Pierce BCA Protein Assay Kit.

**Western blotting**. Samples in soluble fraction were loaded into a 4–20% sodium dodecyl sulfate-polyacrylamide gel (Bio-Rad), and transferred to PVDF Immobilon FL membranes (Millipore). After blocking with 5% non-fat milk in PBS, membranes were blotted overnight with primary antibodies in 5% non-fat milk containing 0.01% Tween-20, and then probed with LI-COR IRDye secondary antibodies or horseradish peroxidase-conjugated secondary antibody, detected by SuperSignal West Femto Chemiluminescent Substrate (Pierce). The information of primary antibodies and their dilutions used in this study are as follows: Cleaved Caspase-3 (Cell Signaling Technology, 9661, 1:1000), Caspase-3 (Cell Signaling Technology, 9662, 1:1000), PSD95 (Abcam, ab2723, 1:1000), Synaptophysin (Abcam, ab8049, 1:1000), AT8 (Thermo Fisher Scientific, MN1020, 1:1000), PHF1 (Abcam, ab184951, 1:1000), Tau5 (Millipore, 577801, 1:1000), APP (Thermo Fisher Scientific, 14-9749-82, 1:1000), GFAP (Millipore, MAB360, 1:1000), EEA1 (Cell Signaling Technology, 2411, 1:1000), LAMP1 (Cell Signaling Technology, 9091, 1:1000), G3BP (BD, 611126, 1:1000), ERCC4 (Fitzgerald, 10R-4026, 1:1000), POLR3A (Abcam, ab96328, 1:1000), and HSPA4 (Cell Signaling Technology, 3303, 1:1000). All uncropped blot images with molecular size markers were available in the Source data file.

**ELISA quantification**. The Aβ40, Aβ42, sAPPα, sAPPβ, CTF-β, and phospho-tau levels in both RIPA and formic acid fractions were measured using the Human β-Amyloid (1–40) ELISA Kit (Thermo Fisher, KHB3481), Human β-Amyloid (1–42) ELISA Kit (Thermo Fisher, KHB3441), Human sAPPα Assay Kit (IBL, 27734), Human sAPPβ Assay Kit (IBL, 27732), Human CTF-β ELISA kit (IBL, 27776), and the Tau (Phospho) [pS396] Human ELISA Kit (Thermo Fisher, KHB7031) according to the manufacturers' instructions. For human apoE ELISA, 96-well plates were coated overnight with an apoE antibody (WUE4) in carbonate buffer at 4 °C. The plates were blocked with 1% milk in PBS, and washed three times with PBS. Recombinant apoE3 and apoE4 (Fitzgerald) were used as standards for the ELISA. Samples were diluted and incubated at 4 °C overnight. The plates were washed and incubated with biotin-conjugated goat anti-apoE antibody (Meridian Life Science) for 2 h at room temperature. After incubation with Horseradish Peroxidase Avidin D (Vector Laboratories) for 90 min at room temperature, the plate was developed by adding tetramethylbenzidine Super Slow substrate (Sigma). The reaction was stopped and read at 450 nm with a microplate reader (Biotek). Results were normalized to total protein concentration of the cell lysate.

**RT-qPCR**. RNA was extracted via the Trizol/chloroform method, followed by DNase and cleanup using the RNase-Free DNase Set and the RNeasy Mini Kit (QIAGEN). The quantity and quality of all RNA samples for RNA-seq was determined by the Agilent 2100 Bioanalyzer using the Agilent RNA 6000 Nano Chip (Agilent Technologies, CA). The cDNA was prepared with the iScript cDNA synthesis kit (Bio-Rad). Real-time qPCR was conducted with Universal SYBR Green Supermix (Bio-Rad) using an iCycler thermocycler (Bio-Rad). The 2exp (−ΔΔCt) method was used to determine the relative expression of each gene with *ACTB* gene coding β-actin as a reference. The primers used to amplify target genes by RT-qPCR are as follows: *ACTB* F (5′-CTGGCACCACACCTTCTACAATG-3′) and R (5′-AATGTCACGCACGATTTCCCGC-3′), *MAP2* F (5′-CAGGTGGCGG ACGTGTGAAAATTGAGAGTG-3′) and R (5′-CACGCTGGA TCTGCCTGGGG ACTGTG-3′), *CTIP2* F (5′-GTTGTGCAAATGTAGCTGGAA-3′) and R (5′-GAA GATGACCACCTGCTCTC-3′), *SATB2* F (5′-CCTTACGCAGAATCTCAGACA A-3′) and R (5′-CCAGATATCTACCAGCAAGTCAG-3′), *GFAP* F (5′-CTGTT GCCAGAGATG GAGGTT-3′) and R (5′-TCATCGCTCAGGAGGTCCTT-3′), *ERCC4* F (5′-TTTGTGAGG AAACTGTATCTG TGG-3′) and R (5′-GTCTGT ATAGCAAGCATGGTAGG-3′), *DGCR8* (PPH17455A, QIAGEN), *POLR3A* F (5′-TCTGGAGACCTGTAGGGACAA-3′) and R (5′-CTGGCTCACCAATGCTC T-3′), *CLP1* (PPH10372A, QIAGEN), *HSPA4* F (5′-GCAACA GCAGCAGACAC CAGC-3′) and R (5′-GCCTTCTTTGGCTTGGGGTGGT-3′), *PNO1* (PPH15310A, QIAGEN), *DND1* F (5′-CTCCACAGGCACCCTGAATG-3′) and R (5′-GGTGCC ATAGGTCCCTGTCC-3′), *NR2F6* F (5′-TTTAGTCGATTTCACGCGGAC-3′) and R (5′-ATCTGTTCAGACGGGTACTCG-3′), *VPS9D1* F (5′-CTCAGACGTCC CCAGGAA CTG-3′) and R (5′-GCCAGACAAGGACAGCTCGTTC-3′), *CTH* F (5′-GAAGACCTACT GGAAGAT-3′) and R (5′-GGAATACTAGCTGTGACT-3′), *CDHR3* F (5′-TGTGAAGGA TGAGGTTGGTG-3′) and R (5′-TCCAGGGTTTG CTCTTTCTAC-3′), *CHIC*1 (PPH09017A, QIAGEN), *ASB3* F (5′-CATACTTAT TTCATCGGGTGC-3′) and R (5′-GGTAACTGCCA ACTGTCCTC-3′), *ARRDC3* F (5′-ATCCCAGTGTGATGTGACGA-3′) and R (5′-TTTGCAA CAGAATCGG AAAA-3′), *TUBB3* F (5′-CCTCCGTGTAGTGACCCTT-3′) and R (5′-GGCCTT TGGACATCTCTTCAG-3′).

**RNA-seq, quality control, and normalization**. Twenty mRNA samples were sequenced at Mayo Clinic using Illumina HiSeq 4000. Reads were mapped to the human reference genome hg38. Raw gene read counts and sequencing quality control were generated using the Mayo Clinic RNA-Seq analytic pipeline: MAP-RSeq Version 3.0[61]. Conditional quantile normalization (CQN) was performed on the raw gene counts to correct for gene length differences, GC bias, global technical variations, and to obtain similar quantile-by-quantile distributions of gene expression levels across samples[62]. Based on the bi-modal distribution of the CQN-normalized and log2-transformed reads per kb per million (RPKM) gene expression values, genes with an average of log2 RPKM > = 3 in at least one group were considered expressed. Using this selection threshold, 18,291 genes were included in the downstream analysis.

**Differential gene expression and pathway analysis**. Differential gene expression analyses were performed using the Partek Genomics Suite (Partek Inc., St. Louis, MO). Gene expression between AD patient- and healthy subject-derived cerebral organoids, between *APOE* genotypes, and the interaction between AD disease status and *APOE* genotype were calculated using analyses of variance models (ANOVA). Pathway analyses of differentially expressed genes, defined by $p < 0.05$ and |fold change| ≥1.5, were performed using MetaCore pathway analysis (© MetaCore (Feb 2020) of Clarivate Analytics. All rights reserved).

**DEGs cell-type distribution analysis of RNA-seq data**. Cell proportion was estimated using marker genes described in BRETIGEA[63], a published reference dataset, and the CIBERSORT program[64]. First, the top 50 marker genes of neuron, astrocyte, and oligodendrocyte cells were obtained from BRETIGEA (version 1.0.0)[63], among which 30 neuronal markers, 27 astrocytic markers, and 19 oligodendrocyte markers are also present in both reference dataset and our iPSC dataset. Next, expression levels of these markers in sorted neurons, astrocytes, and oligodendrocytes were obtained from the reference dataset of Zhang et al.[65]. Finally, CIBERSORT analytical tool (version 1.0.1)[64] was applied to estimate cellular composition in the iPSC dataset using the selected maker gene expression in reference dataset and in the iPSC dataset for neurons, astrocytes, and oligodendrocytes. To estimate in which cell-type genes might be regulated, we applied CellCODE R package[66] to assess the interaction between group variable and surrogate proportion variables of each cell type. Because oligodendrocytes compose negligible proportion of the cell population in all but one sample, oligodendrocyte was not considered in the following analysis. Specifically, getAllSPV function in CellCODE was applied to refine the top 50 markers of neuron and astrocyte obtained from BRETIGEA, and to obtain surrogate variables of neuron and astrocyte populations using the remaining markers through singular value decomposition. Next, cellPopT function was applied to calculate the $t$ statistics of the interaction term between group and the surrogate variables. Cell type with the largest $t$ statistics is the designated cell type.

**WGCNA**. To identify groups of genes that are correlated with AD disease status or with *APOE* genotype, we performed weighted gene co-expression network analysis

(WGCNA)[67] using the log2-transformed, CQN-normalized gene expression values. We used the soft power of 16, hybrid dynamic tree cutting, a minimum module size of 50 genes, and a minimum height for merging modules at 0.3 to build a signed hybrid co-expression networks. Each gene module was summarized by the first principal component of the scaled module expression profiles (module eigengene). Each module was assigned a unique identifier, and genes that did not fulfill these criteria for any of the modules were assigned to the gray module. To assess the correlation of modules to AD disease status and *APOE* genotype, we defined healthy subject-derived cerebral organoids as 0 and AD patient-derived cerebral organoids as 1; and defined the *APOE3* genotype as 0, and the *APOE4* genotype as 1. Modules were annotated using R package anRichment ([https://horvath.genetics.ucla.edu/html/CoexpressionNetwork/GeneAnnotation/](https://horvath.genetics.ucla.edu/html/CoexpressionNetwork/GeneAnnotation/)). The connection among the top hub genes in the yellow module was visualized using VisANT[68].

**Statistical analyses and reproducibility**. ANCOVA for *APOE4*, AD status, sex, sampling age and source of iPSCs was performed to determine the interaction effect between *APOE4* and AD status on each continuous variable for western blotting, ELISA, or RT-qPCR from four groups of cerebral organoids, followed by Tukey–Kramer tests to compare between selected two groups using JMP software version 15.0. For batch difference and isogenic cerebral organoids comparison, the Mann–Whitney U test was performed to determine the significance using GraphPad Prism version 8.0. Spearman correlation analysis was used to analyze the association between proteins using GraphPad Prism. Experiments were repeated in two independent differentiation batches. Data were presented as mean ± SEM. A *p* value of <0.05 was considered statistically significant.

**Reporting summary**. Further information on research design is available in the Nature Research Reporting Summary linked to this article.

## Data availability

Full scans of the gels and blots are available in Source data file. All relevant data are available from the corresponding author upon reasonable request. All the other data supporting the findings of this study are available within the article and its supplementary information files and from the corresponding author upon reasonable request. A reporting summary for this article is available as a Supplementary Information file. The RNA-seq data are available via the AD Knowledge Portal [[https://adknowledgeportal.synapse.org](https://adknowledgeportal.synapse.org)]. The AD Knowledge Portal is a platform for accessing data, analyses, and tools generated by the Accelerating Medicines Partnership (AMP-AD) Target Discovery Program and other National Institute on Aging (NIA)-supported programs to enable open-science practices and accelerate translational learning. The data, analyses, and tools are shared early in the research cycle without a publication embargo on secondary use. Data are available for general research use according to the following requirements for data access and data attribution [[https://adknowledgeportal.synapse.org/DataAccess/Instructions](https://adknowledgeportal.synapse.org/DataAccess/Instructions)]. For access to content described in this manuscript see [[https://doi.org/10.7303/syn22307008](https://doi.org/10.7303/syn22307008)]. Source data are provided with this paper.

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

## Acknowledgements

We are grateful to Dr. Yadong Huang and Chengzhong Wang for providing the APOE isogenic lines. This work was partially supported by Mayo Clinic Center for Regenerative Medicine, Neuroregeneration Lab. Lab Funding: This work was supported by NIH grants RF1AG051504, R37AG027924, RF1AG046205, RF1AG057181, R01AG066395, P01NS074969, and P30AG062677 (to G.B.), Younkin Fellowship and Alzheimer's Association Research Fellowship 2018-AARF-592302 (to J.Z.), and R01AG061796 (to N.E.-T.). Dr. Zbigniew K. Wszolek is partially supported by the Mayo Clinic Center for Regenerative Medicine, the gifts from The Sol Goldman Charitable Trust, the Donald G. and Jodi P. Heeringa Family, the Haworth Family Professorship in Neurodegenerative Diseases fund, and by Albertson Parkinson's Research Foundation.

## Author contributions

J.Z., C.L., F.S., N.E.T., T.K., and G.B. conceived and designed the project, and wrote the paper. M.D., L.J., S.G.Y., N.R.G.R., Z.W., and D.B. helped with collecting human skin biopsies and generating iPSC lines. J.Z., Y.F., Y.Y., W.L., Y.M., K.C., L.J., and T.N. executed the experiments and analyzed the data. Y.R., X.W., Y.C., and Y.A. performed analysis for RNA-sequencing data.

## Competing interests

The authors declare no competing interests.
