## [Peer Review File · Nature Communications]

Reviewers' Comments:

Reviewer #1:

Remarks to the Author:

Zhao et al describe the generation of control and Alzheimer's Disease (AD) cerebral organoids from APOE3/3 or APOE4/4 donors (5 patients each). They compare effects of AD and APOE status to characterize phenotypes that are APOE4 dependent, AD dependent, or co-associated. The authors find several phenotypes that are AD dependent (decreased synaptophysin, PSD95, increased soluble Abeta, etc), APOE4-dependent (ptau accumulation, ApoE protein, etc), and combined APOE and AD status-dependent (apoptosis). The data presented by the authors is very clean (remarkably so, given that 5 independent humans were used for each genotype). The conclusions are reasonable, and further our understanding of the independent and interactive aspects of APOE status and AD. Importantly, the reversal of key phenotypes in an isogenic pair is shown and strongly supports the findings. The paper is lacking, however, in depth of interpretation and contextualization within current literature. These flaws are largely addressable with changes to text. Also, with some of the key findings in this study been previously described in 2D cell culture or organoid models of APOE (e.g. Wang et al, Nature Medicine 2018; Lin et al, Neuron 2018), the most significant novelty rests in the RNA sequencing, which is not examined deeply. With proper revision, the manuscript represents a good candidate for publication in Nature Communications.

Major Comments

The authors need to spend more time describing and analyzing the RNAseq results, and do more to validate by IHC and/or assays. The authors should provide the actual lists of significantly changed genes in the supplement. The authors do not indicate whether overlapping DEGs between conditions are altered in the same direction? Can the authors use a cell-type specific gene expression data (for instance Zhang et al, Neuron 2016) to infer whether DEGs are more from neurons or astrocytes?

Does the WGCNA use the DEGs described in the first paragraph of the section "Altered transcriptional profiles..." or is this an entirely separate analysis? Please show (in supplement perhaps) some validation of the most differentially expressed genes and initial GO analysis.

The authors mention but then never discuss the "magenta" module, which positively correlates with AD. What genes/pathways are enriched in this module? At least one should be validated in their AD organoids.

While most results are consistent with previously published data, some do show distinct outcomes. This is not addressed or discussed in the text. For example, the authors find differences in Abeta load as early as 4 weeks, where Lin and colleagues claimed they couldn't detect any Abeta at 2 months (Lin et al, Neuron 2018). The authors also show higher ApoE protein in APOE4, where Lin and colleagues show lower. Please discuss these differences in the results and/or discussion sections.

Throughout the figures and text, it is unclear exactly which groups are being compared. The authors should clarify the statistics and groups in both figure legends and text. For instance, when the figure states that "APOE4 x AD: $p=0.0057$ " (Fig. 2B), is this comparing Con-E3 to AD-E4 or AD-E3 to AD-E4? Another example relates to Fig 6 where the text states "G3BP significantly increased in organoids from AD patients carrying APOE4 with interaction" but it is unclear from the figure where this significant difference is.

The authors should examine the patterning and GFAP+ content in APOE3 vs APOE4 and control vs AD organoids. If, for example, APOE4 organoids contain significantly more astrocytes, this could explain the ApoE level differences.

Minor Comments

Fig 5A and 5B don't show statistics on the image, but the manuscript text and figure legend suggests that APOE4 x AD is significant.

Fig S1B Sox17/DAPI staining for MC0017 and MC0035 seem of low quality. Are there better images the authors could use?

Reviewer #2:

Remarks to the Author:

This manuscript reports the development and results from a 3-layer cerebral organoid culture from control and AD iPSCs from E3/3 and E4/4 subjects. This paper has several components which will be discussed in sequence.

1. The cerebral organoids. While an n=5 is reported for each of the 4 cohorts, it is potentially significant that 4 of the 5 E4/4-AD patients are female. Females, particularly females with E4, are at an increased risk for AD. That this effects the results at least deserves a convincing discussion. The description of the organoids, comprising Figures 1, 2 and S1 and S2 are excellent. As well, limitations of this system and a description of the next generation is in the Discussion.

2. The neuropathology of the 4 cohorts.

a. The data interpreted as accelerated maturation in AD patients is the result of a decrease in PSD95 and synaptophysin proteins correlated with an increase mRNA from these and other neuronal markers. This conclusion is unclear.

b. The levels of AB40 and AB42 are increased at 8 (Fig. S4) and 12 weeks (Fig. 3) weeks but at not 4 weeks (Fig. S4) with AD in both E3 and E4 organoids, while APP processing does not increase, suggesting impaired clearance. For tau and apoE levels, data is given for both RIPA and FA extraction. There are several issues to be addressed in these data sets. First, a major conclusion of the manuscript is that the increase in tau levels occurs regardless of disease progression and APOE genotype. The only data that demonstrate this assertion is Figure S4B. c. In RIPA and FA extraction buffers, p-tau is significantly greater in E4-AD compared to E3-AD. However, by immunostaining, the PHF antibody staining is negative, suggesting that the positive AT8 data simply indicates the presence of p-tau. For apoE levels, RIPA levels were higher with E4, though higher with AD status in the FA. It should be noted that that RIPA levels are >10-fold higher than FA levels. Indeed, it is discussed that in general, organoids do not exhibit protein deposition, which would be reflected in the FA extraction.

d. The transcriptional profiles are of limited interest as it is unclear how the results included were selected from an immense data set. While amyloid proteins are the top network for APOE4 X AD, the authors seem to focus on stress granule formation, which fits with previous data showing an interaction with RNA binding proteins.

e. An important concept still unclear is whether apoE4 exhibits a loss of positive or gain of negative function. Starting with the title, this manuscript demonstrates only the relationship between E3 and E4. This comparison cannot establish loss of positive or gain of negative function- only a comparison between apoE4 and a lack of apoE can identify this parameter. In the Discussion, the authors incorrectly opine: "Of note, apoE and p-tau levels show positive correlation in both RIPA and FA fractions, which is consistent with the results from mouse studies showing that Apoe deficiency ameliorates tauopathy (42). Thus, APOE4 induces the gain-of-toxic effect on p-tau accumulation in iPSC-derived cerebral organoids, analogous to the observation in iPSC-derived GABAergic neurons (27)."

Reviewer #3:

Remarks to the Author:

In their manuscript "APOE4 exacerbates synaptic loss and neurodegeneration in cerebral organoids from Alzheimer's disease patients", Zhao et al. uses cerebral organoids from iPSCs with APOE

e3/e3 or e4/e4 genotype from individuals with normal cognition or AD to study the role of APOE4 in neurodegeneration and AD pathogenesis. The authors demonstrated that organoids from AD patients showed increased apoptosis, decreased synaptic integrity, increased abeta levels and elevated phosphorylated tau. They then present evidence that APOE4 exacerbates most of these phenotypes. They further highlighted the importance of APOE4 by using isogenically converted APOE3 iPSCs to show that this conversion attenuated the AD related phenotypes. This is a well-written paper providing insight into APOE4 related pathways relevant in AD. However, the interpretation of these findings must be in the context of a fully defined system. Given the known current caveats of organoid technology (variability in differentiation, variability introduced in the necrotic core), a more thorough and transparent description and characterization of their organoid differentiation needs to be provided. If these concerns are addressed, this work would be a valuable contribution to the field. Listed below are the major and minor issues identified by this reviewer.

1. The authors must describe the heterogeneity in their differentiations across organoids and differentiation rounds in more in depth, and explain if and how this variability was taken into account in their analysis.
 - a. First, the authors need to state how many rounds of differentiations the data are coming from, and the variability across different rounds.
 - b. Second, were there exclusion criteria based on size, or composition of an organoid? If so, what percentage of the organoids failed to pass?
 - c. Finally, it is essential to provide information on percentages of different cell types present across organoids within a differentiation and across differentiation rounds. This can be demonstrated by immunostaining and quantification. In parallel, western blot, qPCR and a more thorough presentation of the RNAseq data for multiple markers of neurons and astrocytes should be included.
2. While the authors demonstrated that organoids from AD patients show higher levels of abeta, phospho-tau, and cleaved caspase3 than organoids from individuals with normal cognition, they need to include more normalization methods in order to take into account the differentiation efficiency. They need to show whether or not these findings are due to technical differences in differentiation efficiency or due to diagnosis and APOE genotype.
3. As mentioned by the authors in the discussion section, one caveat of the RNAseq analysis is that "the core of cerebral organoids show necrosis-like changes after 3 months of differentiation, likely due to the lack of vascular systems". The authors need to first define variability in this necrotic core, then explore whether the necrotic core, rather than APOE genotype or AD status, could be driving the observed differences in apoptosis and cleaved caspase.
4. In figure 6, the authors only show graphs with top GO terms and 50 genes of their RNAseq data. They need to state where their RNAseq data will be deposited and shared.
5. In figure 6E, the G3BP immunostaining is not convincing. In order to more effectively portray the punctate staining, the authors should show images with higher magnification.
6. In figure S4, the order of groups in figure legend doesn't match the order of groups in the graph.
7. In figure 2B, the data points for AD-E4 are very dispersed. Could the significance be driven by organoids from one particular line? Addressing the above mentioned differentiation heterogeneity issue would help address this.
8. In some figures, the authors use n=25 (5 organoids per line, then 5 lines per group) and in others, they use N=5. Does that mean N=5 is using 1 organoid per group? Or consider average of

5 organoids as N=1? More explanation is required in the figure legends.

May 3, 2020

Re: NCOMMS-19-42071

Reviewer #1

Zhao et al describe the generation of control and Alzheimer's disease (AD) cerebral organoids from APOE3/3 or APOE4/4 donors (5 patients each). They compare effects of AD and APOE status to characterize phenotypes that are APOE4 dependent, AD dependent, or co-associated. The authors find several phenotypes that are AD dependent (decreased synaptophysin, PSD95, increased soluble Aβ, etc), APOE4-dependent (p-tau accumulation, ApoE protein, etc), and combined APOE and AD status-dependent (apoptosis). The data presented by the authors is very clean (remarkably so, given that 5 independent humans were used for each genotype). The conclusions are reasonable, and further our understanding of the independent and interactive aspects of APOE status and AD. Importantly, the reversal of key phenotypes in an isogenic pair is shown and strongly supports the findings. The paper is lacking, however, in depth of interpretation and contextualization within current literature. These flaws are largely addressable with changes to text. Also, with some of the key findings in this study being previously described in 2D cell culture or organoid models of APOE (e.g. Wang et al, Nature Medicine 2018; Lin et al, Neuron 2018), the most significant novelty rests in the RNA sequencing, which is not examined deeply. With proper revision, the manuscript represents a good candidate for publication in Nature Communications.

--- We thank the reviewer for detailed review and helpful comments.

Major Comments

The authors should provide the actual lists of significantly changed genes in the supplement. The authors

do not indicate whether overlapping DEGs between conditions are altered in the same direction? Can the authors use a cell-type specific gene expression data (for instance Zhang et al, Neuron 2016) to infer whether DEGs are more from neurons or astrocytes?

---We thank the reviewer for valuable suggestions and comments. We have added the lists of significantly changed genes as the supplement data-significant altered gene list, in which the directions of DEGs were shown as positive (+) or negative (-) fold changes. From the results, we found that most overlapping DEGs are changed in the same direction. Among 302 overlapping genes between Con-E3 vs. Con-E4 and AD-E3 vs. AD-E4, 265 genes were changed in the same direction. Among 317 overlapping genes between AD-E3 vs. Con-E3 and AD-E4 vs. Con-E4, 304 genes were changed in the same direction. We have included the information in page 9, line 23 – page 10, line 5. In addition, to assess the cell type preference of DEGs, we analyzed the RNA-seq data using CIBERSORT and CellCODE programs. First, cell proportion was estimated using marker genes described in BRETIGEA, a published reference dataset, and the CIBERSORT program. To estimate which cell type correlates the best with group differences beyond its cellular composition fluctuation, we applied the CellCODE R package to assign the best cell type to each DEG. Because oligodendrocytes compose negligible proportion of the cell population, it was not considered in this analysis. The analysis revealed that cerebral organoids were mainly composed of neurons (median: 0.9231) and astrocytes (median: 0.06698). No significant differences in neuron and astrocyte proportion were observed among different groups. We also found that more DEGs were designated to neurons than those assigned to astrocytes, indicating neurons contributed more to the differences of cerebral organoids from different groups. We have now included these new data in Fig. S7 and described the findings in page 10, line 16- page 11, line 2. The detailed analytic method is described in the Methods section (page 25, line 22-page 26, line 17).

Does the WGCNA use the DEGs described in the first paragraph of the section “Altered transcriptional profiles...” or is this an entirely separate analysis? Please show (in supplement perhaps) some validation of the most differentially expressed genes and initial GO analysis.

---We thank the reviewer for valuable suggestions and comments. The WGCNA was performed for all the transcriptome data, and was not based on DEGs. Nonetheless, to ensure the consistency in the quality control pipeline, we have re-performed the DEG and pathway analyses in the supplement Figure S6A-G and revised the descriptions in page 10, lines 7-12. We have also validated the most differentially expressed genes via RT-qPCR as shown in the supplement Figure S6H-L and Page 10, lines 13-15.

The authors mention but then never discuss the “magenta” module, which positively correlates with AD. What genes/pathways are enriched in this module? At least one should be validated in their AD organoids.

---We appreciate the reviewer’s valuable comments. We have now added the information for the “magenta” module, which is closely related to DNA and nucleosome metabolism pathways. Selective hub genes were also validated via RT-qPCR. The new data are now included in the supplement Figure S8 and described in the main text (page 11, lines 10-14).

While most results are consistent with previously published data, some do show distinct outcomes. This is not addressed or discussed in the text. For example, the authors find differences in Abeta load as early as 4 weeks, where Lin and colleagues claimed they couldn’t detect any Abeta at 2 months (Lin et al, Neuron 2018). The authors also show higher ApoE protein in APOE4, where Lin and colleagues show lower. Please discuss these differences in the results and/or discussion sections.

---We appreciate the reviewer's valuable comments. We should emphasize the major differences in experimental procedures to measure the amounts of AD-related molecules in cerebral organoids. For example, while Lin et al used Western blotting and failed to detect A β at early stage (2 months old), we used more sensitive ELISA to measure A β . We have added the descriptions in page 8, lines 4-6. More importantly, we used a different protocol from that reported by Lin et al to differentiate human iPSCs into cerebral organoids, though the principles used in these protocols are similar. The differences in the work flows and reagents might potentially influence the properties of cerebral organoids. For example, Lin et al add 10% FBS in the medium to maintain the cerebral organoid culture, while we cultured the cerebral organoids under serum-free condition, which could contribute to the different results on apoE levels by the two studies. Depending on the components of lipids, cytokines and/or growth factors in culture medium, apoE production in cerebral organoids may be differently modulated under the stress conditions induced by *APOE4* and/or AD status. We have discussed these points in page 17, line 20 – page 18, line 2.

Throughout the figures and text, it is unclear exactly which groups are being compared. The authors should clarify the statistics and groups in both figure legends and text. For instance, when the figure states that “*APOE4* x AD: $p=0.0057$ ” (Fig. 2B), is this comparing Con-E3 to AD-E4 or AD-E3 to AD-E4? Another example relates to Fig 6 where the text states “G3BP significantly increased in organoids from AD patients carrying *APOE4* with interaction” but it is unclear from the figure where this significant difference is.

---We thank the reviewer's important comments. In our analysis, *APOE4* and AD status were used as two independent variables to determine their interaction effect (*APOE4* x AD) on a continuous variable. The P value labeled in figures indicated the significance of the difference between selected two groups analyzed by Tukey-Kramer tests among four groups. Only the statistically significant comparisons by Tukey-Kramer tests were shown in the figures. We have now modified the description in figure legends and Methods section.

The authors should examine the patterning and GFAP+ content in *APOE3* vs *APOE4* and control vs AD organoids. If, for example, *APOE4* organoids contain significantly more astrocytes, this could explain the ApoE level differences.

---We thank the reviewer for the valuable suggestions and comments. Accordingly, we have analyzed the levels of GFAP through immunostaining, RT-qPCR and Western blotting. We did not observe any significant effects of *APOE4*, AD or *APOE4* x AD on GFAP levels after normalized by those for Tuj1. Thus, *APOE4* is predicted to increase apoE levels without increasing astrocyte population in the organoids. We have included the new data as the supplement Figure S5, and described in page 9, lines 9-12.

Minor Comments

Fig 5A and 5B don't show statistics on the image, but the manuscript text and figure legend suggest that *APOE4* x AD is significant.

---As described above, *APOE4* and AD status were used as two independent variables to determine their interaction effect (*APOE4* x AD) on a continuous variable. Our results indicate that *APOE4* influences apoE levels in the RIPA fraction (Fig. 5A), while AD status significantly influences apoE levels in the FA fraction (Fig. 5B). However, interaction between *APOE4* and AD (*APOE4* x AD) was not significant in both cases. The P values labeled in the figures indicated the significance of the difference between

selected two groups analyzed by Tukey-Kramer tests. Since any comparisons between two groups did not show statistically significant differences by Tukey-Kramer tests in the analyses, statistics was not shown in Fig. 5A and B.

Fig S1B Sox17/DAPI staining for MC0017 and MC0035 seem of low quality. Are there better images the authors could use?

---Thanks for pointing this out. We have changed the images with higher quality in the supplement Figure S1.

Reviewer #2:

This manuscript reports the development and results from a 3-layer cerebral organoid culture from control and AD iPSCs from E3/3 and E4/4 subjects. This paper has several components which will be discussed in sequence.

--- We thank the reviewer for detailed review and helpful comments.

1. The cerebral organoids. While an n=5 is reported for each of the 4 cohorts, it is potentially significant that 4 of the 5 E4/4-AD patients are female. Females, particularly females with E4, are at an increased risk for AD. That this effects the results at least deserves a convincing discussion. The description of the organoids, comprising Figures 1, 2 and S1 and S2 are excellent. As well, limitations of this system and a description of the next generation is in the Discussion.

---We agree with the reviewer on the unmatched sex distribution in our study. Although we adjusted sex in our analysis, we recognize the limitation of these studies using patient-derived iPSC models on addressing potential sex-dependent effects. We have included additional discussions in page 18, lines 3-5.

2. The neuropathology of the 4 cohorts.

a. The data interpreted as accelerated maturation in AD patients is the result of a decrease in PSD95 and synaptophysin proteins correlated with an increase mRNA from these and other neuronal markers. This conclusion is unclear.

---At Week 12, *APOE4* and AD status synergically reduced the amounts of synaptic proteins PSD95 and synaptophysin in the cerebral organoids as shown in Fig. 2E and F. In contrast, we found that an increase of PSD95 and synaptophysin proteins accompanied with increased mRNA levels of mature neuronal markers at the early stage of differentiation (Week 4) in AD-organoids compared to control organoids as shown in the supplement Figure S3. Thus, we speculate that extreme neuronal differentiation/maturation is synergistically induced by AD-related factors and *APOE4* in the pre-symptomatic stage, and that the mechanistic exhaustion followed by neuronal dysfunction may contribute to disease development. We have discussed this point in page 16, lines 13-16.

b. The levels of AB40 and AB42 are increased at 8 (Fig. S4) and 12 weeks (Fig. 3) weeks but at not 4 weeks (Fig. S4) with AD in both E3 and E4 organoids, while APP processing does not increase, suggesting impaired clearance. For tau and apoE levels, data is given for both RIPA and FA extraction. There are several issues to be addressed in these data sets. First, a major conclusion of the manuscript is

that the increase in tau levels occurs regardless of disease progression and APOE genotype. The only data that demonstrate this assertion is Figure S4B.

--- Our results indicate that *APOE4* exacerbates tau pathology in both control and AD organoids at week 12, while A β and p-tau levels were higher in AD organoids than control organoids. We have modified the description in Abstract and Introduction.

c. In RIPA and FA extraction buffers, p-tau is significantly greater in E4-AD compared to E3-AD. However, by immunostaining, the PHF antibody staining is negative, suggesting that the positive AT8 data simply indicates the presence of p-tau. For apoE levels, RIPA levels were higher with E4, though higher with AD status in the FA. It should be noted that that RIPA levels are >10-fold higher than FA levels. Indeed, it is discussed that in general, organoids do not exhibit protein deposition, which would be reflected in the FA extraction.

---In Figure 4B-D, p-tau levels in the RIPA lysates were analyzed by Western blotting using AT8 (Ser202/Thr205) antibody and PHF1 (Ser396/Ser404) antibody, which were normalized to total tau levels. We did not detect any differences in total tau levels among the 4 groups. As described in the Methods section, we performed sequential two-step extraction using RIPA and FA buffers to analyze detergent soluble fraction (RIPA) and insoluble fraction (FA). Thus, our results in Figure 5 indicate that more apoE exists in soluble fraction than insoluble fraction in the cerebral organoids.

d. The transcriptional profiles are of limited interest as it is unclear how the results included were selected from an immense data set. While amyloid proteins are the top network for APOE4 X AD, the authors seem to focus on stress granule formation, which fits with previous data showing an interaction with RNA binding proteins.

---We thank the reviewer's comments. In DEGs with significant *APOE4* x AD interaction, "amyloid proteins" was identified as a top-ranked pathway. To further explore gene networks which modulated by *APOE4* and/or AD status through an unbiased approach, we conducted WGCNA in the dataset from RNA-seq.

e. An important concept still unclear is whether apoE4 exhibits a loss of positive or gain of negative function. Starting with the title, this manuscript demonstrates only the relationship between E3 and E4. This comparison cannot establish loss of positive or gain of negative function-only a comparison between apoE4 and a lack of apoE can identify this parameter. In the Discussion, the authors incorrectly opine: "Of note, apoE and p-tau levels show positive correlation in both RIPA and FA fractions, which is consistent with the results from mouse studies showing that Apoe deficiency ameliorates tauopathy (42). Thus, APOE4 induces the gain-of-toxic effect on p-tau accumulation in iPSC-derived cerebral organoids, analogous to the observation in iPSC-derived GABAergic neurons (27)."

---We agree with the reviewer that our study cannot lead the conclusion on whether *APOE4* exhibits a loss of positive or gain of negative function. Nonetheless, Wang et al have shown that p-tau levels were increased in APOE-deficient iPSC-derived neurons when treated with lysates of iPSC-derived neurons with APOE4, suggesting *APOE4* may induce the gain-of-toxic effect on p-tau accumulation. Thus, we have modified those descriptions in page 15, line 17-20.

Reviewer #3:

In their manuscript “APOE4 exacerbates synaptic loss and neurodegeneration in cerebral organoids from Alzheimer’s disease patients”, Zhao et al. uses cerebral organoids from iPSCs with APOE e3/e3 or e4/e4 genotype from individuals with normal cognition or AD to study the role of APOE4 in neurodegeneration and AD pathogenesis. The authors demonstrated that organoids from AD patients showed increased apoptosis, decreased synaptic integrity, increased abeta levels and elevated phosphorylated tau. They then present evidence that APOE4 exacerbates most of these phenotypes. They further highlighted the importance of APOE4 by using isogenically converted APOE3 iPSCs to show that this conversion attenuated the AD related phenotypes. This is a well-written paper providing insight into APOE4 related pathways relevant in AD. However, the interpretation of these findings must be in the context of a fully defined system. Given the known current caveats of organoid technology (variability in differentiation, variability introduced in the necrotic core), a more thorough and transparent description and characterization of their organoid differentiation needs to be provided. If these concerns are addressed, this work would be a valuable contribution to the field. Listed below are the major and minor issues identified by this reviewer.

--- We appreciate the reviewer for detailed review and valuable comments.

1. The authors must describe the heterogeneity in their differentiations across organoids and differentiation rounds in more in depth, and explain if and how this variability was taken into account in their analysis.

a. First, the authors need to state how many rounds of differentiations the data are coming from and the variability across different rounds.

--- We thank the reviewer for the comments. In our study, we differentiated 20 iPSC lines into cerebral organoids within a close time frame. Thus, our results shown in the manuscript came from one round of differentiation. We have stated this point in page 20, lines 20-22.

b. Second, were there exclusion criteria based on size, or composition of an organoid? If so, what percentage of the organoids failed to pass?

---During the differentiation process, abnormal cerebral organoids fell apart into small pieces in the orbital shaker, which were mostly eliminated during medium change. We collected all cerebral organoids with normal size range (typically 2-4 mm in diameter) and excluded those with obviously smaller sizes (< 1mm). We have stated this point in page 20, lines 20-22. We found approximately 1-2 abnormal organoids in each dish (Fig. I, for reviewers’ references). There was no evident difference in the ratio of abnormal organoids from different groups.

Figure I. Representative cerebral organoids at Week 12. Cerebral organoids with normal size were collected for further experiments. The one that was much smaller than others was excluded (red arrow).

c. Finally, it is essential to provide information on percentages of different cell types present across organoids within a differentiation and across differentiation rounds. This can be demonstrated by immunostaining and quantification. In parallel, western blot, qPCR and a more thorough presentation of the RNAseq data for multiple markers of neurons and astrocytes should be included.

--- We thank the reviewer for the comments. Since we are aware of the importance in assessing percentages of different cell types in cerebral organoids, we do plan to conduct single cell RNA-sequencing to measure cellular compositions in each cerebral organoid in our future studies. Alternatively, we have investigated the levels of astrocytic marker GFAP and neuronal marker Tuj1 through immunostaining, RT-qPCR and Western blot. Since we did not observe any significant effects of *APOE4*, AD or *APOE4* x AD on the ratio of GFAP/Tuj1, *APOE4* and AD status unlikely influence cellular composition of cerebral organoids. We have included the new data as the supplement Figure S5, and described in page 9, lines 9-12. Furthermore, we also analyzed GFAP and Tuj1 levels in the new cerebral organoids prepared through the second round of differentiation. We found no significant differences in the expression of GFAP and Tuj1 at Week 12 in both first and second round experiments (Fig. S9). These have been described in page 6, line 14-18. To estimate the cellular composition fluctuation, we also analyzed the RNA-seq data using CIBERSORT and CellCODE program. While oligodendrocytes compose negligible proportion of the cell population, we found that cerebral organoids were mainly composed of neurons (median: 0.9231) and astrocytes (median: 0.06698). We have now included these new data in Fig. S7 and described the findings in page 10, lines 16- page 11, line 2. The detailed analytical method is described in the Methods section (page 25, line 22-page 26, line 17).

2. While the authors demonstrated that organoids from AD patients show higher levels of abeta, phospho-tau, and cleaved caspase3 than organoids from individuals with normal cognition, they need to include more normalization methods in order to take into account the differentiation efficiency. They need to show whether or not these findings are due to technical differences in differentiation efficiency or due to diagnosis and APOE genotype.

---We agree with the reviewer for the importance in exploring better methods to normalize those measurements by considering the differentiation efficiency in cerebral organoids. However, variations of those methods might require tremendous efforts, which are beyond scope of our current study but are planned in our future studies. Nonetheless, we have investigated the levels of GFAP and Tuj1, and showed that *APOE4* and AD status do not influence cellular composition of cerebral organoids in the supplement Figure S5. To address the reviewer's concern on the influence of technical differences to the results, we have repeated several key experiments using the new cerebral organoids prepared through the second round of differentiation. We have confirmed consistent results to those from the first round shown in the main figures; 1) Cerebral organoids from AD patients carrying *APOE* $\epsilon 4/\epsilon 4$ have greater apoptosis; 2) cerebral organoids from AD patients have increased levels of $A\beta$ and p-tau, and 3) *APOE4* exacerbates tau pathology in both control and AD organoids (Fig. II, for reviewers' references). Together, these results indicate that effects of technical differences on experimental outcome are relatively small to impact our main conclusions.

Figure II. *APOE4* exacerbates AD-related pathologies in cerebral organoids from AD patients (Second round experiments). Cerebra organoids were collected for immunostaining and ELISA at week 12. (A) Representative images of cellular apoptosis evaluated by immunostaining of cleaved CASP3. Scale bar: 50 μ m. (B) Quantifications of cleaved CASP3 immunoreactivity (*APOE4*: $p=0.0008$, AD: $p=0.0556$, *APOE4* x AD: $p=0.0041$). (C-E) Amounts of A β 40 (C; *APOE4*: $p=0.6572$, AD: $p=0.0017$, *APOE4* x AD: $p=0.5629$) and A β 42 (D; *APOE4*: $p=0.1331$, AD: $p=0.0009$, *APOE4* x AD: $p=0.8152$) in the RIPA fraction were measured by ELISA. The ratio of A β 42/A β 40 was calculated accordingly (E; *APOE4*: $p=0.2585$, AD: $p=0.062$, *APOE4* x AD: $p=0.8882$). (F, G) Amounts of p-tau in the RIPA fraction (F; *APOE4*: $p=0.0002$, AD: $p=0.0004$, *APOE4* x AD: $p=0.3805$) and the FA fraction (G; *APOE4*: $p=0.0001$, AD: $p=0.0003$, *APOE4* x AD: $p=0.9682$) were measured by ELISA. Data were normalized to individual total protein concentration. All data are expressed as mean \pm SEM (N=5). ANCOVA for *APOE4*, AD status, sex, sampling age, and source of iPSCs was performed to determine the interaction effect between *APOE4* and AD status on each variable, which was followed by Tukey-Kramer tests to compare between selected two groups among four groups. * $p<0.05$, ** $p<0.01$, *** $p<0.001$, **** $p<0.0001$.

3. As mentioned by the authors in the discussion section, one caveat of the RNAseq analysis is that “the core of cerebral organoids show necrosis-like changes after 3 months of differentiation, likely due to the lack of vascular systems”. The authors need to first define variability in this necrotic core, then explore whether the necrotic core, rather than APOE genotype or AD status, could be driving the observed differences in apoptosis and cleaved caspase.

---The gradual presence of necrotic core in the center of cerebral organoids is one of the major caveats of this model system due to limited availability of nutrients. Since the necrotic core becomes macroscopically detectable after Week 12 in our models, we did not use organoids cultured beyond Week 12 for experiments to minimize the effect of necrotic core. Nonetheless, we agree with the reviewer that necrotic core may potentially influence cleaved caspase levels. Thus, the immunoreactivity of cleaved CASP3 was measured only in the surface neuronal layers in Fig. 2B according to the published method by Gonzalez et al. (Mol Psychiatry, 2018: 23, 2363-2374). To further address the reviewer’s concern, we evaluated the level of cleaved CASP3 in the center and the edge of the cerebral organoids at Week 12 from the AD-*APOE4/4* line and its *APOE3/3* isogenic line by immunostaining (Figure S10) with results described in Page 12, line 11-16. As expected, the center of the cerebral organoids showed higher immunoreactivity of cleaved CASP3 than the edge. We observed no significant difference in the center of cerebral organoids from the parental line and isogenic line, whereas there is the significant increase at the edge of the cerebral organoids from the AD-*APOE4/4* parental line compared to those from isogenic *APOE3/3* line. Correlation analysis showed no significant correlation between the cleaved CASP3 level at

the center and the edge, suggesting the apoptosis at the edge of cerebral organoids is not driven by the necrotic core at the center.

4. In figure 6, the authors only show graphs with top GO terms and 50 genes of their RNAseq data. They need to state where their RNAseq data will be deposited and shared.

--- The RNA-seq data are available via the AD Knowledge Portal

(<https://adknowledgeportal.synapse.org>). The AD Knowledge Portal is a platform for accessing data, analyses, and tools generated by the Accelerating Medicines Partnership (AMP-AD) Target Discovery Program and other National Institute on Aging (NIA)-supported programs to enable open-science practices and accelerate translational learning. The data, analyses and tools are shared early in the research cycle without a publication embargo on secondary use. Data is available for general research use according to the following requirements for data access and data attribution (<https://adknowledgeportal.synapse.org/DataAccess/Instructions>).

See the following links for direct data access:

- Gene expression data: <https://doi.org/10.7303/syn22005901>
- Processed data: <https://doi.org/10.7303/syn22005904>

For additional information and metadata on this study, see the following link:

<https://adknowledgeportal.synapse.org/Explore/Studies?Study=syn21680862>.

5. In figure 6E, the G3BP immunostaining is not convincing. In order to more effectively portray the punctate staining, the authors should show images with higher magnification.

---Thank you for the suggestion. We have changed the previous images with images of higher magnification in Figure 6E.

6. In figure S4, the order of groups in figure legend doesn't match the order of groups in the graph.

---Thank you for pointing this out. We have corrected it in the Figure S4.

7. In figure 2B, the data points for AD-E4 are very dispersed. Could the significance be driven by organoids from one particular line? Addressing the above mentioned differentiation heterogeneity issue would help address this.

---We appreciate the reviewer's comments. We have re-analyzed the data by averaging the measurements of 5 organoids from one independent iPSC line in the revised manuscript. However, we did not find any specific outliers standing out in the dataset.

8. In some figures, the authors use $n=25$ (5 organoids per line, then 5 lines per group) and in others, they use $N=5$. Does that mean $N=5$ is using 1 organoid per group? Or consider average of 5 organoids as $N=1$? More explanation is required in the figure legends.

---For all of the WB and ELISA analysis, we pooled and homogenized 4-5 cerebral organoid from one independent iPSC line as 1 sample. Thus, the data represents results from 5 independent iPSC lines per group. In the original version of manuscript, we compared measurements of 5 organoids/line from 5 iPSC lines ($n=25$) for the immunostaining analysis (Figure 2B) and size comparison (Figure S2). To avoid any confusion, we have now re-analyzed the data by averaging the measurements of 5 organoids from one independent iPSC line and comparing the values from 5 independent iPSC lines per group as $N=5$ /group. We have now revised the graphs and figure legends in Figure 2B and Figure S2.

We trust that we have sufficiently revised our manuscript and responded all reviewers' suggestions and comments. We hope that our revised manuscript is now acceptable for publication in the Nature Communications.

Sincerely,

Guojun Bu, Ph.D.

Mary Lowell Leary Professor and Chair

Reviewers' Comments:

Reviewer #1:

Remarks to the Author:

The authors have largely addressed the critiques raised, including using computational methods to suggest from which cell types the DEGs arise in the organoids, which extends the interpretations possible. Most of our requested control quantifications were performed to satisfaction. The analysis of the RNAseq data is still minimal which could be deepened, but the revisions made by the authors satisfy the requests made.

One remaining concern is that the authors validate their most significant changed genes by RT-qPCR, which serves as a technical validation but not a great biological one. The authors could have validated a few of their most significant genes with an orthogonal method such as RNAscope, immunohistochemistry or western.

Reviewer #3:

Remarks to the Author:

Zhao et al. have addressed concerns to a satisfactory level, and this revised version is a good candidate for publication in Nature Communications.

Reviewer #1 (Remarks to the Author):

The authors have largely addressed the critiques raised, including using computational methods to suggest from which cell types the DEGs arise in the organoids, which extends the interpretations possible. Most of our requested control quantifications were performed to satisfaction. The analysis of the RNAseq data is still minimal which could be deepened, but the revisions made by the authors satisfy the requests made.

One remaining concern is that the authors validate their most significant changed genes by RT-qPCR, which serves as a technical validation but not a great biological one. The authors could have validated a few of their most significant genes with an orthogonal method such as RNAscope, immunohistochemistry or western.

Response: We agree with the reviewer that the RNAseq data is valuable and the analysis could be deepened. To further validate the RNA-seq results in a biological way, we focused on the genes significantly changed in the yellow module. Protein levels of selective hub genes (ERCC4, POLR3A and HSP4A) were evaluated by Western blotting. Consistent with the RT-qPCR results, the protein levels were also decreased in AD-E4 organoids. We have now included these new data in Fig. 6F, J-L and described the findings in page 11, lines 16-17.

Reviewer #3 (Remarks to the Author):

Zhao et al. have addressed concerns to a satisfactory level, and this revised version is a good candidate for publication in Nature Communications.

Response: We sincerely appreciate the insightful comments and support from the reviewer.

We trust that we have sufficiently revised our manuscript and responded all reviewers' suggestions and comments. We hope that our revised manuscript is now acceptable for publication in the Nature Communications.

Sincerely,

Guojun Bu, Ph.D.

Mary Lowell Leary Professor and Chair